# Exploring dynamic interactions of single nanoparticles at interfaces for surface-confined electrochemical behavior and size measurement

Hui Ma[1,2,5], Jian-Fu Chen[3,5], Hai-Feng Wang[3], Pei-Jun Hu[3,4], Wei Ma[1✉] & Yi-Tao Long[1,2✉]

With the development of new instruments and methodologies, the highly dynamic behaviors of nanoparticle at the liquid-solid interface have been studied. However, the dynamic nature of the electrochemical behavior of individual nanoparticles on the electrode interface is still poorly understood. Here, we generalize scaling relations to predict nanoparticle-electrode interactions by examining the adsorption energy of nanoparticles at an ultramicroelectrode interface. Based on the theoretical predictions, we investigate the interaction-modulated dynamic electrochemical behaviors for the oxidation of individual Ag nanoparticles. Typically, significantly distinct current traces are observed owing to the adsorption-mediated motion of Ag nanoparticles. Inspired by restraining the stochastic paths of particles in the vicinity of the electrode interface to produce surface-confined current traces, we successfully realize high-resolution size measurements of Ag nanoparticles in mixed-sample systems. This work offers a better understanding of dynamic interactions of nanoparticles at the electrochemical interface and displays highly valuable applications of single-entity electrochemistry.

[1] School of Chemistry and Molecular Engineering, East China University of Science and Technology, 200237 Shanghai, P. R. China. [2] State Key Laboratory of Analytical Chemistry for Life Science, School of Chemistry and Chemical Engineering, Nanjing University, 210023 Nanjing, P.R. China. [3] State Key Laboratory of Chemical Engineering, Centre for Computational Chemistry & Research Institute of Industrial Catalysis, East China University of Science and Technology, 200237 Shanghai, P. R. China. [4] School of Chemistry and Chemical Engineering, The Queen's University of Belfast, Belfast BT9 5AG, UK. [5] These authors contributed equally: Hui Ma, Jian-Fu Chen. ✉email: weima@ecust.edu.cn; yitaolong@nju.edu.cn

Single-entity electrochemistry has attracted increasing attention not only for its extremely high sensitivity but also for the insight that it offers with respect to individual direct electron transfer that is masked in ensemble-averaged measurements[1–3]. Over the past several decades, electrochemical measurements of individual entities, including nanoparticles (NPs), cells, molecules, and atoms, have been achieved conveniently with the use of stochastic collision measurement at a ultramicroelectrode (UME) or nanoelectrode[4–6]. In a typical collision event, individual diffusing entities stochastically collide with a UME interface where they may adsorb, directly react, or enable a catalytic reaction[7–9]. The information yielded has usually been in the form of staircase or blip responses in individual current signals, corresponding to NPs that either have irreversibly adsorbed onto or rebounded off the UME surface[10–12]. Significant advances have been made in the analysis of the dynamic electrochemical behaviors of individual entities[13–15]. For example, studies of single AgNP dynamics on UMEs, beginning with Compton's work, have been extensively pursued over the past several years[16]. Recently, complicated multiple spikes in the current response were observed for the electrooxidation of a single AgNP using high-resolution electrochemical instruments[17,18]. These experiments have been spearheaded by Oja et al.[13], Ustarroz et al.[14] and our group[15]. This distinct phenomenon was explained by the fact that AgNP partially oxidized in one stochastic collision and tended to oxidize incrementally during the repeated multicollision processes at the solution−electrode interface[13]. Although the combination of stochastic collision electrochemistry with scanned nanoprobe microscopies[19,20] or optical-based techniques[21] was introduced as a new tool for real-time analysis of the electrochemical reaction mechanism/kinetics of individual NPs with high spatial and temporal resolution[22,23], it remains an open and challenging issue to elucidate how nanoscale particles interact with an electrode interface to clearly understand the dynamic electrochemical processes of such particles[24–26].

In a single-NP collision electrochemical measurement, the random walk behavior of NPs, that is Brownian motion, results in the collision of NPs with a UME[27,28]. Freely diffusing particles can move differently as NPs approach the interface of the UME due to the boundary effect[15,29]. Typically, surface diffusion coefficients are often orders of magnitude lower than those in bulk systems because the NP-interface interactions significantly affect the interfacial diffusion of NPs[30,31]. One of the fundamental challenges underlying particle motion is to understand the dynamic forces that shape these interactions. Available experimental evidence has demonstrated a dominant mechanism of desorption-mediated NP motion at the liquid−solid interface with a transient weakly adsorbed state between desorption and adsorption[32–35]. Thus, adsorptive interactions play a key role in the physical motion of NPs at the electrode interface. Recent electrochemical results provided a preliminary insight on the physicochemical interactions between NPs and electrode surfaces[14], yet clearly clarifying the dependence of NP−electrode interactions and the dynamic nature of single-NP electrochemical behavior remains unexplored. Here, using first-principle density functional theory (DFT) calculations, we study the adsorption-energy scaling relations by examining NPs on two different UME surfaces. This allows us to semiquantitatively predict the adsorptive interaction effect of NPs on UME interfaces and helps in identifying the dynamic electrochemical nature of single NPs during the stochastic collision process. Based on the theoretical predictions, we explore the interaction-modulated electrochemical oxidation behavior of individual AgNPs, producing significantly distinguished current traces. By attempting to suppress the stochastic path of AgNPs and control their motion in an alkaline media, we observe the complete oxidation of the AgNPs with different diameters of 10, 18, 34, 55, 65 and 75 nm, and successfully realize the in situ size discrimination of AgNP mixtures with multiparameter analysis using the stochastic collision electrochemistry.

## Results

**Prediction of NPs−electrode interactions.** Typical interactions between NPs and the electrode interface in an electrolyte involve electrostatic, van der Waals, solvation, solvophobic and depletion forces[36–38]. It has also been shown that the electronic structure of nanomaterials is strongly correlated with their adsorption properties and that these, in turn, determine the interaction between the NPs and an interface[39]. By this way, adsorption energy has been demonstrated to be a key semiquantitative parameter for describing the strength of the interaction of NPs with a UME interface[15].

Relevant research has shown that adsorptive scaling relations can be generalized to include the local structure of the adsorption site[39,40]. To date, AgNPs are among the most commonly used particles for studying the direct electrochemical behaviors of individual entities[39,41]. Thus, we studied the adsorption properties of surface terminations of AgNPs and silver oxide ($AgO_x$) NPs on two of the most commonly used UME materials in nanoelectrochemistry: carbon fiber UMEs (C UMEs) and Au UMEs (Fig. 1a). The simplicity of the concept of coordination number ($n_c$) and its widespread use in computational chemistry make this parameter as a possible descriptor to introduce the sensitivity of geometry and structure into scaling relations. For the description of particles, scaling relations were obtained between the AgNP radius $r$ and the average coordination $n_c$ ($n_c = 9 − 3d_{Ag}/r$, where $d_{Ag}$ is the bond length between two Ag atoms; see details in Supplementary Note 1), determined by approximating the AgNPs as icosahedral entities with 20 orderly arranged triangular-type (111) facets (Supplementary Table 1 and Supplementary Fig. 1)[15]. To estimate the adsorption energy for an NP surface morphology with multiple terrace, edge, and corner sites, different $n_c$ of the surface Ag atoms were ingeniously constructed by adding Ag atoms to the original Ag (111) surface[15]. While the added Ag atoms were replaced by O atoms on AgNP surface to approximate a description of $AgO_x$ NPs (see Supplementary Fig. 2). Thus, the total adsorption energy ($E_{ad}^{tot}$) of an NP from DFT calculations can be described by Eq. (1):

$$E_{ad}^{tot} = E_{NP/electrode} − E_{electrode} − E_{NP}, \qquad (1)$$

where $E_{NP/electrode}$, $E_{electrode}$, and $E_{NP}$ are the energies of the NP adsorbed on the electrode interface, the electrode, and the NP, respectively. $E_{ad}$ of atoms with different $n_c$ were determined from $E_{ad}^{tot}$ by averaging the surface adatoms. A series of possible adsorption sites on the Ag or O atom of the carbon (C, 002) basal plane and Au (111) surface were systematically checked. Essentially, we carefully constructed active sites with $n_c$ between 3 and 9. The most energetically stable adsorption configurations under study are shown in Fig. 1b (top view, denoted as Ag/C, Ag/Au, and $AgO_x$/Au) and Supplementary Figs. 3–5 (side view). In these systems, each Ag or O atom is free to uprightly adsorb on the C (002) basal plane and Au (111) surface with a different generalized $n_c$. Supplementary Table 2 shows the DFT-calculated $E_{ad}$ and $n_c$ for the Ag/C, Ag/Au, and $AgO_x$/Au systems. Figure 1c shows the scaling relations between the adsorption energies of AgNP or $AgO_x$ NP on the C and Au surface facets. The strong correlation between $E_{ad}$ and $n_c$ at the surface adsorption sites is evident and suggests that the corresponding adsorption energy becomes more negative when the surface coordination $n_c$ decreases, exhibiting a good dependence from 3 to 9 (Fig. 1c).

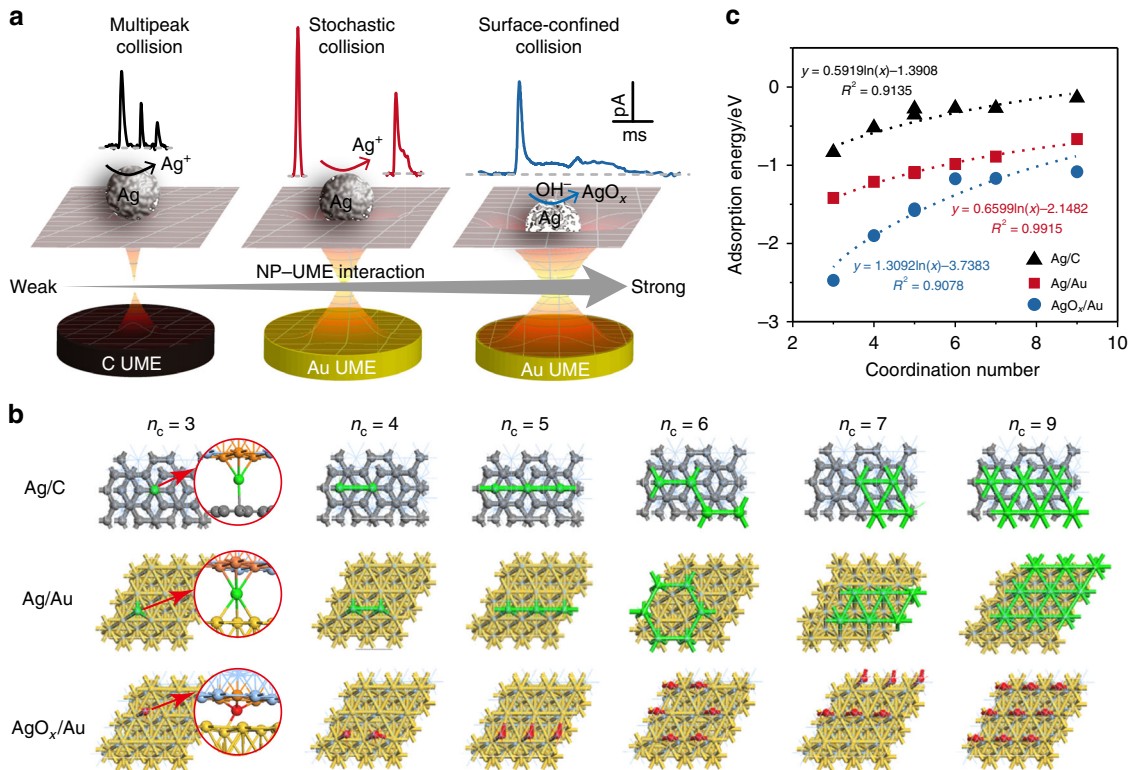

**Fig. 1 Modeling NPs—electrode interactions. a** Schematic illustration of interaction-modulated dynamic electrochemical behaviors of single NPs. **b** Optimized most stable adsorption configurations of Ag atoms and O atoms with different coordination environments on the C (002) basal plane and the Au (111) surface for DFT calculation. Gray ball, red ball, green ball/blue line and yellow ball correspond to C, O, Ag and Au atoms, respectively. **c** Scaling relations in adsorption energy of NPs on C and Au UME surfaces as a function of generalized coordination number. Data for Ag/C (black triangle), Ag/Au (red square) and AgO$_x$/Au (blue circle) are provided, respectively. Source data are provided as a Source Data file.

This correlation is approximately logarithmic for certain ranges of coordination (typically $n_c \geq 3$) and can be well fitted by $E_{ad} = a \ln n_c + b$, where $a = 0.59$ and $b = -1.39$ for Ag/C; $a = 0.66$ and $b = -2.15$ for Ag/Au; and $a = 1.31$ and $b = -3.74$ for AgO$_x$/Au, respectively. The more negative $E_{ad}$ is, the more strongly atoms adsorb to the electrode interface. On the basis of adsorptive scalability, the coordination dependence of the adsorption energies was found to vary in the order AgO$_x$/Au > Ag/Au > Ag/C, which indicates significantly attractive interactions in the AgO$_x$/Au system due to the strongest adsorption of the NP —electrode interface. The scaling relations between adsorption energies can directly describe interactions and accurately capture the adsorptive trends of NPs on the surface of UMEs. Thus, these systems are expected to exhibit distinct dynamic motion behaviors as a result of their different adsorptive interaction strengths.

**Interaction-modulated electrooxidation of single AgNPs.** Inspired by the theoretical predictions, we can establish an experimental framework to investigate the interaction-modulated electrochemical behaviors of individual AgNPs (Fig. 1a). Using a low-noise electrochemical measurement platform, time-resolved current traces were directly observed for the dissolution of individual AgNPs at +600 mV vs. Ag/AgCl wire (Fig. 2a). As expected, 34 nm AgNPs afforded typical electrochemical oxidation at an Au UME in a 25 mM PB solution at pH = 7.4, displaying two major current patterns arising from a single large peak and a spike with undulating terrain (Fig. 2d). In this case, a single peak with a height of 582.6 ± 32.9 pA (gray bar, Supplementary Fig. 6a) and a dwell time of 0.37 ± 0.01 ms (gray bar, Fig. 2g) corresponds to a single collision event on the Au UME surface. A spike with undulating terrain, showing a smaller current amplitude of 213.6 ± 26.8 pA (red bar, Supplementary Fig. 6a) and a longer duration of 1.05 ± 0.07 ms (red bar, Fig. 2g), suggests that the AgNP fluctuates in the tunneling region of the electrode interface. Mixed motion is produced due to a through-space mild adsorptive interaction with AgNPs approaching the electrode interface, matching two current traces. As a result, the two current patterns exhibit different charge ranges within a usual variation (right, Fig. 2g). Moreover, the size distributions of the 34 nm AgNPs obtained from the integrated charge slightly deviated from the size analysis (Supplementary Fig. 7), especially for the large single-peak signals, suggesting the incomplete dissolution of AgNPs.

For the weak adsorption of the Ag/C system, we also examined the electrochemical oxidation of AgNPs, in which the working electrode was replaced by a C UME in a 25 mM PB solution at pH = 7.4 (Fig. 2b). A spike with closely spaced clustering was clearly predominant for 34 nm AgNPs (Fig. 2e); the amplitude of the first spike decreased to 65.8 ± 7.1 pA (black bar, Supplementary Fig. 6b), while the duration increased to 2.80 ± 0.55 ms (left, Fig. 2h). This current feature occurred because the partially oxidized AgNPs receded to the bulk solution and returned to the C UME surface several times due to the weak adsorption between the AgNPs and C UME, resulting in a transient weakly bound state between adsorption and desorption. Poisson distribution model demonstrates the consecutive current spikes can rather be attributed to the oxidation of a single AgNP via a series of stages, during each of which the NP is partially oxidized (see details in Supplementary Note 2); thus, the electrooxidation of single AgNPs

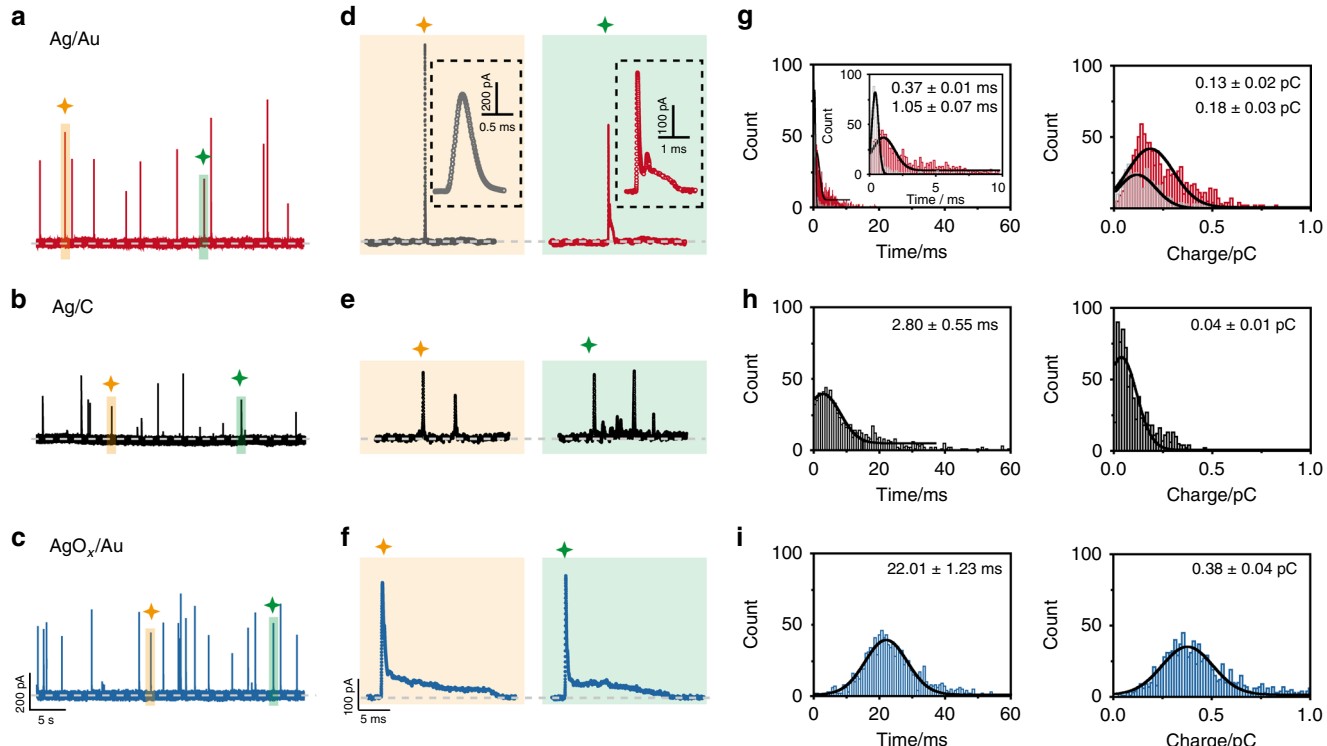

**Fig. 2 Interaction-modulated dynamic electrochemical oxidation behaviors of single AgNPs.** Current responses of individual 34 nm AgNPs collisions at +600 mV vs. Ag/AgCl **a** in neutral solution (25 mM PB, pH = 7.4, red line) at an Au UME (diameter 12.5 μm), **b** in neutral solution (25 mM PB, pH = 7.4, black line) at a C UME (diameter 7 μm), and **c** in alkaline media (15 mM PB and 10 mM NaOH, pH = 11.4, blue line) at an Au UME (diameter 12.5 μm). **a–c** Chronoamperometric profiles for the electrochemical oxidation of individual AgNPs with the same scale bar. **d–f** Close-ups of the representative time-resolved current traces, corresponding to the spikes in **a–c**. Inset: zoom-in of the corresponding current traces. **g–i** Histograms showing the distributions of the durations and charges of AgNPs. Black curves show Gaussian fits. The data were obtained from a large population of electrooxidation events of individual AgNPs (more than 1000 events). Source data are provided as a Source Data file.

experiences a longer timescale in this case. Moreover, this behavior means AgNPs have a higher escape probability from the C UME surface and do not return, causing significantly incomplete oxidation. This conclusion is supported by the measured results from the integrated charge associated with each collision event being much smaller than the charge expected based on the AgNP size (right, Fig. 2h). In addition, an unpredictably spaced timescale in multicollision processes was observed, which could also be attributed to the stochastic nature of the weak adsorption of AgNPs at the C UME surface.

To further demonstrate the effect of adsorptive interaction-modulated electrochemical behavior, we attempted to investigate the strong coupling of the AgO$_x$/Au system using stochastic collision measurements. Considering the formation of silver oxide in alkaline media[42–47], we performed the electrochemical oxidation of 34 nm AgNP at an Au UME in a pH = 11.4 of an alkaline solution containing 15 mM PB and 10 mM NaOH (Fig. 2c). Notably, we observed a series of current transients, which is more than 90% of cases showed a spike with undulating terrain (Fig. 2f) with a current amplitude of 290.3 ± 12.9 pA (blue bar, Supplementary Fig. 6c) and a very long duration of 22.01 ± 1.23 ms (left, Fig. 2i). Nearly uniform current traces were observed; that is, particles always fluctuated in the tunneling region at a timescale of tens of milliseconds. And the dwell time of particles in alkaline media increased to 20 times that in neutral solution, which was attributed to the slow electrooxidation kinetics because of the insoluble oxide layer on the surface of silver[22,23,48]. Considering the participation of OH⁻, we assumed the oxidation reaction of AgNPs as a diffusion-limited process in alkaline media. The

maximum current ($i_{max}$) of a single AgNP on the surface of the electrode can be described by Eq. (2) for a sphere NP[11]:

$$i_{max} = 4\pi nFD_0 C_0 r_0 \ln 2, \quad (2)$$

where $n$ is the number of electrons transferred per Ag atom, $D_0$ is the diffusion coefficient of OH⁻ ($6.8 \times 10^{-9}$ m² s⁻¹)[49], $C_0$ is the concentration of OH⁻, $r_0$ is the radius of AgNPs. For an AgNP with a diameter of 34 nm, the expected diffusion-limited current ($i_{max}$) is 491.1 pA in alkaline media (pH = 11.4), which is much higher than the average maximum current of 290.3 pA in our experimental measurement (Supplementary Fig. 6c). This would suggest the concentration of OH⁻ in our system is sufficient for the transition from Ag⁺ to AgO. We inferred that the diffusion of OH⁻ to the AgNPs was not the rate-determining step of the AgNPs oxidation, evidencing that the electrooxidation of AgNPs is the reaction rate-limited processes. Thus, the elongated current signal was mainly attributed to the formation of the insoluble silver oxide on the particle surface, resulting in the slow oxidation kinetics of AgNPs in alkaline media. Moreover, the enhanced attractive interaction between the AgO$_x$ NPs and the Au UME suppressed the random walk of particles during the electrochemical stripping process to produce uniform current traces and promote the complete oxidation of AgNPs in alkaline media. Interestingly, the integrated charge of individual AgNPs was almost twice that of neutral solution (right, Fig. 2i); this relationship probably resulted from the 2e⁻ oxidation of Ag to form the AgO product. To confirm the 2e⁻ process, we carried out differential pulse voltammetry (DPV) measurements using an AgNP-modified Au electrode (diameter 2 mm) in alkaline media

with pH = 11.4. Four anodic peaks, referred to $A_1$–$A_4$ in the DPV results, coincide with those reported in previous reports and are associated with the formation of $Ag_2O$ and AgO between −200 mV and +800 mV vs. Ag/AgCl (Supplementary Fig. 8 and Supplementary Note 3)[44–46], and no obvious oxidation peak could be found in the cyclic voltammogram for a bare Au electrode in the absence of AgNPs (Supplementary Fig. 9). Additionally, the integrated charge of individual collision events as a function of applied potential (Supplementary Fig. 10), the X-ray photoelectron spectroscopy and X-ray diffraction results for the electrochemically oxidized product (Supplementary Fig. 11 and Supplementary Note 4) further demonstrated the formation of AgO at +600 mV vs. Ag/AgCl wire[50–52]. Therefore, a 2e⁻ oxidation occurred in the process of Ag oxidation when holding the potential of +600 mV vs. Ag/AgCl wire in alkaline media. In analogy to pitting corrosion, we proposed that the AgNP oxidation in alkaline solution followed a heterogeneous nucleation growth-type mechanism by the stochastic formation of a nucleus of silver oxide on the surface of AgNPs[43–45]. According to the previous work, the presence of NaOH preceded a reaction of $Ag^+$ ions, resulting from the oxidation of AgNPs, with $OH^-$ ions to allow the formation of $[Ag(OH)_2]^-$ for further oxidation to produce silver oxide[44]. The higher formation constants for the silver oxides likely make the initiation process more favorable, resulting in the maximum current at the first spike due to the fast rate of initial nucleation. After initiation, the subsequent slow rates of AgNPs oxidation, as reflected in the resulting low undulating terrain shapes, may either arise from the formation of the insoluble silver oxide on the particle surface or alternately may relate to the dynamic motion of the particle adjacent to the electrode interface.

**Electrochemical size measurement of individual AgNPs**. The uncontrolled dynamic motion and partial oxidation of AgNPs create an obstacle to size measurement using stochastic collision electrochemistry[12,53]. Inspired by confining the stochastic paths of particles in the vicinity of the Au UME surface to produce uniform current traces in alkaline solution, a surface-confined nano-collision strategy was introduced for the electrochemical size measurement of AgNPs in a liquid suspension. To choose the favorable experimental condition for this surface-confined nano-collision, pH effect of single AgNPs in the oxidation process was investigated using the stochastic collision electrochemistry. As shown in Supplementary Fig. 12 and Supplementary Table 3, prolonged surface-confined current traces and 2e⁻ transfer process were observed when pH was beyond 11.4, which was sufficient for the oxidation of AgNP to form AgO (see details in Supplementary Note 5). Hence, unless otherwise stated, the pH value of the electrolyte solution at 11.4 was used. In this study, an Au UME with a diameter of 12.5 μm and an applied potential of +600 mV vs. Ag/AgCl wire was employed to attempt the electrochemical measurement of individual AgNPs with different diameters of 10, 18, 34, 55 and 65 nm in pH = 11.4 alkaline media containing 15 mM PB and 10 mM NaOH. The representative current signals illustrated that asymmetric exponential current transients initially rise quickly and decay at a lower rate with some fluctuation for the oxidation of individual AgNPs, except for a single peak for 10 nm diameter AgNPs due to their small size and fast oxidation (Fig. 3a). Moreover, the magnitude and duration of current peaks increased dramatically as the size of AgNPs increased; these peaks were accompanied by uniform current traces, indicating well-distinguishable size-dependent characteristic current traces. Based on these measured current signals, we further investigated the average numbers of oxidized Ag atomic layers during the first subpeak of individual collision

events (Supplementary Fig. 13 and Supplementary Note 6) and modeled the AgO film growth kinetics on the surface of different sized AgNPs using a dynamic Monte Carlo simulation (Supplementary Fig. 14). We found a size-dependent critical AgO thickness for the transition from fast to slow oxidation, which grows exponentially with the increasing diameter of AgNPs.

Considering 2e⁻ oxidation in alkaline media at the potential of +600 mV vs. Ag/AgCl wire, we estimated the size distribution from the integrated charge of the oxidation of AgNPs by Eq. (3)[25]:

$$d_{NP} = 2\sqrt[3]{\frac{3QAr}{4\pi nF\rho}}, \tag{3}$$

where $d_{NP}$ is the diameter of the AgNP, $Q$ is the integrated charge from the current transient, Ar is the atomic molecular mass of Ag (107.9 g mol⁻¹), $n$ is the number of electrons transferred per Ag atom, F is Faraday's constant and $\rho$ ($10.5 \times 10^6$ g m⁻³) is the density of Ag. As shown in Fig. 3b, the mean diameter values obtained from the electrochemical measurements are 8.6, 15.7, 33.5, 51.4 and 62.8 nm, closely resembling those determined via transmission electron microscopy (TEM) (Fig. 3c and Supplementary Fig. 15). When individual pure suspensions of AgNPs with the five different sizes in alkaline solution (pH = 11.4) were measured via dynamic light scattering (DLS), significant overlap and wide peak widths were observed in the compiled size distribution diagrams (Fig. 3d). To further demonstrate this surface-confined nano-collision strategy, we also examined the oxidation behaviors of 75 nm AgNPs in alkaline solution (Supplementary Fig. 16). With an analysis of current responses, a statistical size distribution can be obtained, and a reliable correlation between TEM and DLS can be constructed. We attributed the remarkable sensitivity of size discrimination in single-NP electrochemical detection to the enhanced adsorptive interaction between the particles and UME, which confined the stochastic motion and prompted the complete oxidation of the AgNPs. Our method pushes the detection limit for stochastic collision measurements from the previous detection limit of ∼30 nm diameter citrate-capped AgNPs in nitrate-based electrolyte[12] and ∼50 nm diameter of AgNPs in the presence of halide ions[45,54,55] up to ∼75 nm AgNPs in the present study. Moreover, these excellently resolved current traces yielded clearly distinguishable histograms of Log $t$ and Log $i$ using a python-based self-designed software program within several minutes (see details on the data processing in Supplementary Fig. 17 and Supplementary Note 7). Gaussian peak values of Log $t$ and Log $i$ from the analysis of over 1000 collisions of each sample were assigned to NPs of various sizes with high sensitivity to realize the size discrimination of AgNPs from 10 to 65 nm in diameter (Supplementary Fig. 18 and Supplementary Note 8).

By virtue of the exceptional sensitivity achieved in the size detection of single AgNPs, we then examined the resolving power to measure a mixture of AgNPs in four sizes ranging from 10 to 55 nm in diameter in situ. As expected, distinguishable spike signals were recorded at an Au UME at a potential of +600 mV vs. Ag/AgCl wire. Four representative traces revealed that the current heights clustered around four different amplitude levels, which corresponded to four differently sized AgNP populations (Fig. 4a). To further demonstrate the sensitivity of the surface-confined nano-collision strategy, AgNPs with average sizes of 34, 55, 10, and 18 nm were added and detected in one electrochemical cell in sequence. Because the collision frequency of AgNPs can approach 50−60 events per second, statistically representative histograms of the dimension, Log $t$ and Log $i$ was obtained by rapidly interrogating thousands of NPs individually at a time interval of a few minutes using the home-built software (Fig. 4b−e). By multiparameter analysis, accurate dimension distributions of the AgNP mixture were achieved. The median

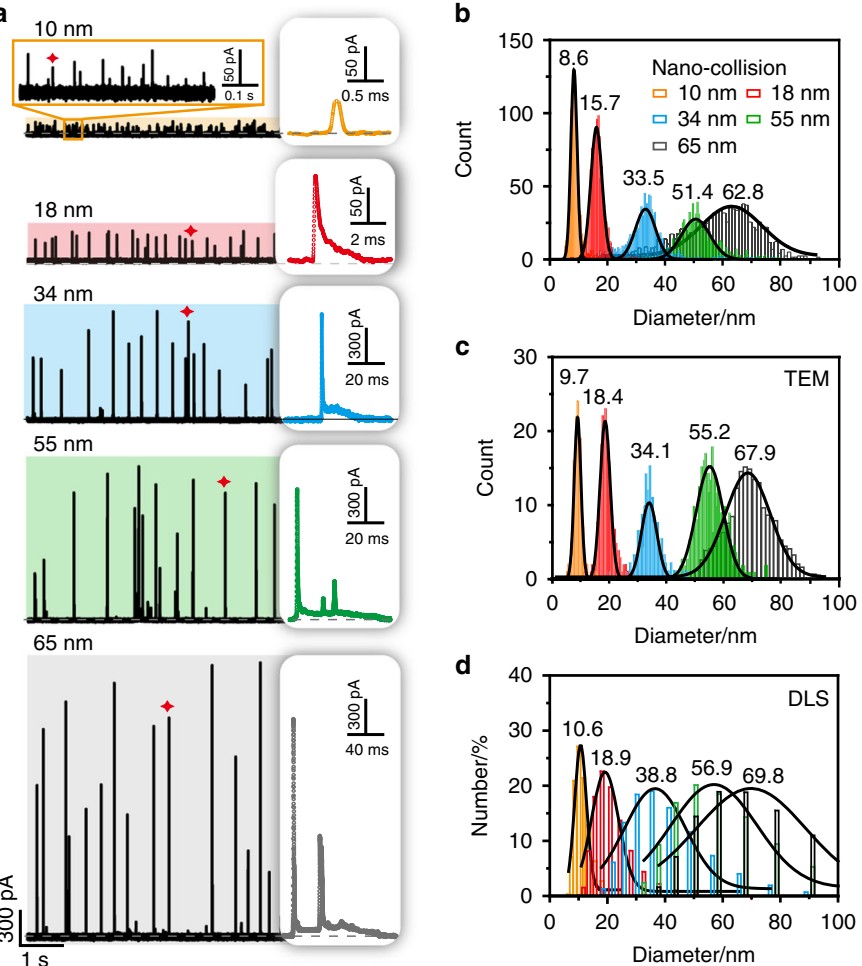

**Fig. 3 Electrochemical size measurement of individual AgNPs by surface-confined nano-collision. a** Chronoamperometric curves and representative current signals of individual AgNPs with different sizes in diameters of 10, 18, 34, 55 and 65 nm at +600 mV vs. Ag/AgCl wire in alkaline solution at an Au UME (diameter 12.5 μm), respectively. Size distribution histograms of AgNPs for individual samples of five sizes of NPs measured separately via nano-collision (**b**), TEM (**c**) and DLS (**d**). Source data are provided as a Source Data file.

sizes (ca. 8.9, 15.7, 32.0 and 52.7 nm) measured in this fashion are in good agreement with the results of the electrochemical detection (Fig. 3b) and TEM measurement (Fig. 3c) of pure suspensions of AgNPs with four different sizes, and the corresponding distributions are more statistically representative than those obtained via the other methods[13,16]. To further confirm the attribution of these distributions, AgNPs with sizes of 34 and 18 nm in diameter were injected successively. An obvious increase in the distribution of the diameter, Log $t$ and Log $i$ was observed for these two sizes (Fig. 4f, g). All results indicated that the surface-confined nano-collision strategy could accurately measure the size of AgNP mixtures in liquid suspensions in situ.

## Discussion
In summary, we explored the utility of theoretical investigation by DFT calculations for predicting the adsorption properties of NPs on the UME interface, which could be used as a descriptor of NP−electrode interactions. We showed that the continuous monitoring of the electrochemical oxidation of individual AgNPs in different adsorptive systems produced significantly distinguished current traces due to the interaction-mediated motion behaviors of the AgNPs. By enhancing the adsorptive interaction between NPs and the electrode interface, the stochastic motion of NPs was restricted to produce surface-confined current traces for the complete dissolution and oxidation of individual AgNPs. This

allowed us to realize high-resolution size measurements of AgNPs mixtures with different sizes in liquid suspensions using stochastic collision electrochemistry, leading to an effective method for particle size analysis in a complex environment. Through this study, we have revealed the implicit role that the adsorptive interaction of NPs on the electrode interface plays in particle motion. In the previously obtained result, the electrostatic force is also an important interaction between AgNPs and electrode interface at sufficiently positive potentials (≥1.2 V vs. Pt wire)[22]. Therefore, it is impossible to describe with certainty all the relevant interactions of NPs on an electrochemical interface, but the current pockets of adsorption knowledge provide a conceptual framework to guide this exploration. In the next step, we will consider the generality and applicability of the surface-confined nano-collision strategy and explore the possible extension to other particles (metallic, polymeric, macromolecules, etc.) at the electrode interface.

## Methods

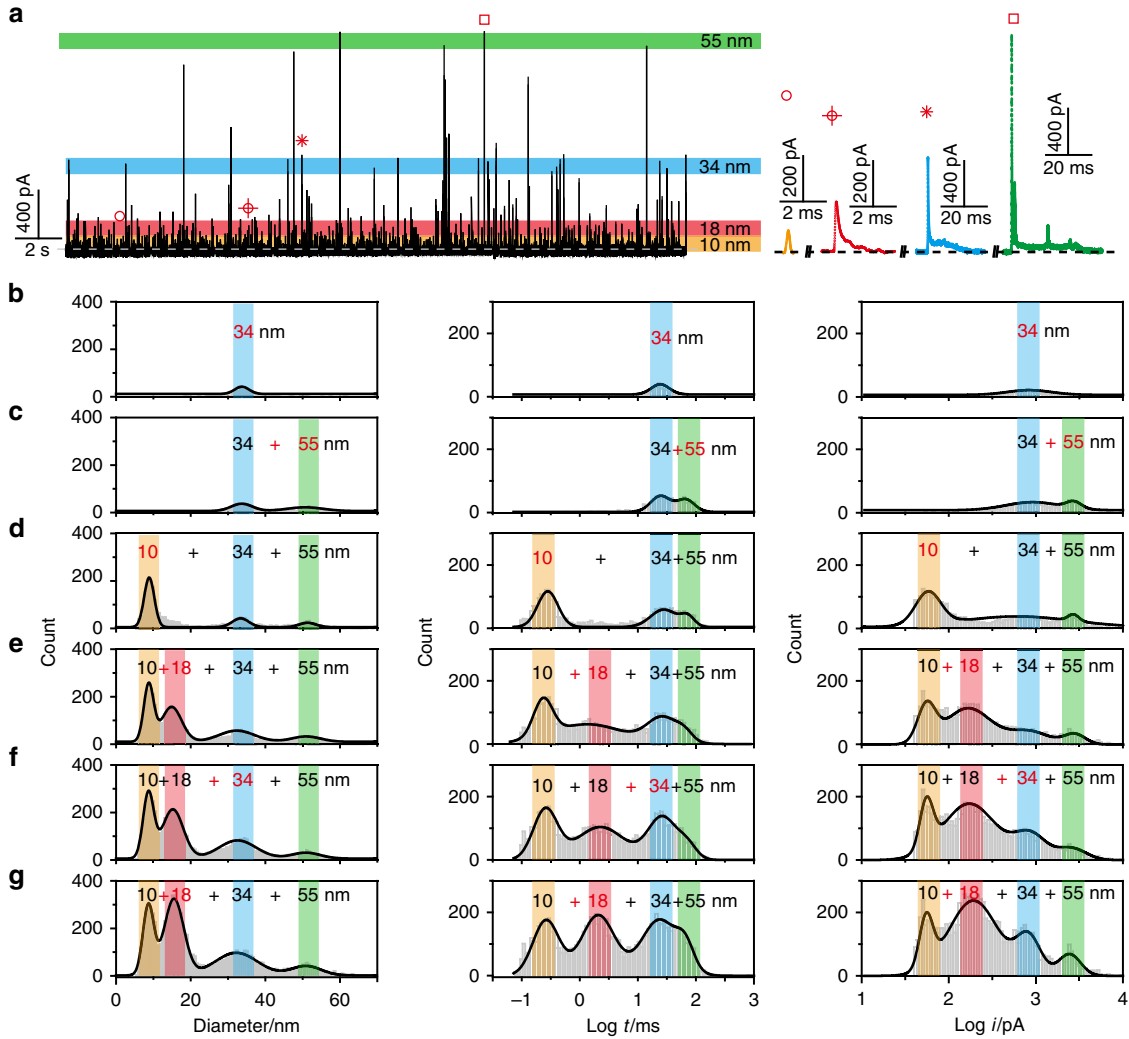

**Fig. 4 In situ electrochemical size measurement of different sized AgNPs mixture. a** Continuous chronoamperometric curves and characteristic current traces for a mixture of AgNPs with different sizes in diameters of 10, 18, 34, 55 nm, respectively. The colored bands represent the maximum current distribution for each analyte in the AgNPs mixtures. **b**–**g** Histograms of the diameter, Log $t$ and Log $i$ for a mixture of AgNPs. The data were acquired from AgNPs with difference sizes, which were added in one electrochemical cell in sequence (in bold red font), at an Au UME (diameter 12.5 μm) in 15 mM PB and 10 mM NaOH at pH = 11.4 and +600 mV vs. Ag/AgCl wire. Source data are provided as a Source Data file.

solutions. Phosphate buffer solution (PB) was prepared using $Na_2HPO_4$ and $NaH_2PO_4$. The electrolyte solutions were prepared with Millipore water.

**Instrumentation**. Chronoamperometric curves were performed using low-noise current measurement platform with the two-electrode cell system (a UME and an Ag/AgCl wire reference electrode). Au electrodes and Pt reference electrodes were purchased from Shanghai Chenhua Co., Ltd., China. Axopatch 200B amplifier and DigiData 1550A low-noise data acquisition system (Molecular Devices, USA) were employed to recode the high-resolution electrochemical signals of individual AgNPs at a sampling rate of 100 kHz and an internal low-pass Bessel filter of 5 kHz. The pH value of a solution was measured by a PHS-3C pH meter (Switzerland Mettler Toledo Delta 320).

**DFT calculations**. All the spin-polarized calculations were performed using a Perdew–Burke–Ernzerhof functional within the generalized gradient approximation using the VASP code[56–58] and optB88 functional with the self-consistent vdW interaction included[59,60]. The project-augmented wave method was used to represent the core-valence electron interaction[61,62]. To model the carbon electrode and Au electrode, a four-layer $p(2 \times 2\sqrt{3})$ C (002) basal plane $p(3 \times 3)$ and Au (111) surface slab corresponding to 64 C atoms and 36 Au atoms were constructed with a $3 \times 3 \times 1$ $k$-point sampling, respectively, and a vacuum layer of 25 Å was applied along the $z$-direction to avoid interactions between periodically repeated slabs. C $2s^2 2p^2$, Au $5d^{10}6s^1$ and Ag $4d^{10}5s^1$ electrons were treated as valence electrons. For total energy calculations, the valence electronic states were expanded in plane wave basis sets with an energy cutoff of 500 eV, and the force convergence criterion in the structure was set to 0.05 eV Å$^{-1}$. The bottom two layers of the substrates were

fixed, and the other atoms were all relaxed during the structural optimization. To simulate the interface solution environment, the solvent effects were in combination with the polarizable continuum model[63,64]. To investigate the interactions between NPs and electrode interface, first the three-layer Ag (111) surfaces of Ag particle with $p(\sqrt{3} \times 3)$ and $p(3 \times 3)$ were constructed to match the C (002) and Au (111) surface as much as possible, respectively (1.7 and 2.2% lattice mismatches, respectively); then, by adding Ag atoms to the original Ag (111) surface, the surface Ag atoms were ingeniously constructed with completely equivalent coordination environments of 3-, 4-, 5-, 6-, 7-, 9-fold, respectively (Supplementary Figs. 3 and 4). While the added Ag atoms were replaced by O atoms to Ag (111) surface in order to approximate the description of $AgO_x$ NPs (Supplementary Fig. 5). The different configurations of the fivefold coordination are also tested and show similar adsorption energies, demonstrating the reliability of our strategy (Supplementary Table 2 and Supplementary Figs. 4 and 5). The calculation model of Ag/Au is taken from the previous work[15,65]. Considering the negligible contribution, we ignore the effects of electrolyte (Supplementary Table 4), charge (Supplementary Table 5 and Supplementary Fig. 19) and capping agent (Supplementary Table 6) to theoretical energy calculations in this DFT model (see details in Supplementary Note 9).

**Electrochemical measurement**. For DPV measurements of AgNPs with different sizes of 10, 18, 34 and 55 nm, an Au electrode (2 mm in diameter) was used as the working electrode, Ag/AgCl as the reference electrode while a Pt wire electrode acted as a counter electrode (at 25 °C), with the solutions degassed with $N_2$ prior to use. All working electrodes were polished before the experiments with 0.3 and 0.05 μm alumina powder on a polishing pad. Then, the polished Au electrodes were reversibly cycled in a 0.5 mM $H_2SO_4$ solution from 0 to +1.55 V (vs. saturated

calomel electrode) until an ideal redox wave of $H_2SO_4$ was observed. The cleaned electrode was thoroughly rinsed with water and absolute ethanol and then was immersed in Ag nanoparticle suspension for 24 h, then the electrode was left in nitrogen for about 20 min to evaporate the solvent.

**Single nanoparticle collision experiments**. Stochastic collision experiments were performed using a low-noise current measurement platform with the two-electrode cell system. The amplifier's internal low-pass Bessel filter was set as 5 kHz. Data were acquired at a sampling rate of 100 kHz. The electrochemical cell consisted of a UME and an Ag/AgCl wire reference electrode. All single-entity electrochemical measurements were performed in a Faraday cage at 25 °C.

**Characterization of AgNPs**. Transmission electron microscopy images were obtained using a JEOL 20100F transmission electron microscope (JEOL Ltd., Japan). Sample preparation consisted of drop casting AgNPs suspensions on holey carbon grids and allowing to dry. AgNPs of each size were then sized using Gatan Digital Micrograph software. DLS analysis was performed with a Zetasizer Nano-ZS Instrument (ZEM4228) from Malvern Instruments in alkaline solution (15 mM PB and 10 mM NaOH, pH 11.4). X-ray photoelectron spectroscopy and X-ray diffraction were recorded by an ESCALAB 250 Xi spectrometer (Thermo Fisher, USA) and a D/max 2550 VB/PC diffractometer (Rigaku, Japan), respectively.

## Data availability

The data that support the findings of this study are available from the corresponding authors upon reasonable request. The source data underlying Figs. 1–4 and Supplementary Figs. 3−14, 16, 18 and 19 are provided as a Source Data file. All other data are available from the corresponding authors upon reasonable request.

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

## Acknowledgements

This research was supported by the National Natural Science Foundation of China (21775043), Shanghai Municipal Natural Science Fund (19ZR1472100), Shanghai Pujiang Program (2019PJD010), the Research Foundation for Talented Scholars of Nanjing University (14912224), the Program of Introducing Talents of Discipline to Universities (B16017), the Fundamental Research Funds for the Central Universities (222201718001, 222201717003) and the China Postdoctoral Science Foundation (2019TQ0143, 2019M660108).

## Author contributions

W.M. and Y.-T.L. conceived and managed the project. H.M., W.M. and Y.-T.L. designed the experiments. H.M. performed the experiments, analyzed the data and wrote the manuscript. J.-F.C. performed the DFT simulation and co-wrote the manuscript. H.-F.W. and P.-J.H. performed the DFT simulation. W.M. analyzed the data, wrote the manuscript and organized the experimental and simulated results. All authors discussed the results and commented on the manuscript.

## Competing interests

The authors declare no competing interests.
