## [Peer Review File · Nature Communications]

Reviewers' Comments:

Reviewer #1:

Remarks to the Author:

This is a high quality paper, detailing a thorough experimental and simulation based approach to considering the oxidation of Ag nanoparticles during electrochemical collisions. The data is well presented and analysed appropriately and inferences/interpretations are made based on literature, DFT simulation which are reasonable and logical.

This work is a valuable addition to a rapidly expanding field, and builds on the earlier body of literature nicely.

I am happy to recommend publication in present form, albeit a reference to justify the applicability of a polar continuum model for solvent within the DFT calculations at this length scale would be welcomed in the text.

Reviewer #2:

Remarks to the Author:

This study contributes to the "sizing" of silver nanoparticles as a particular model system in single entity electrochemistry. This work also contributes to the discovery of surface phenomena that contribute to the nanoparticle - surface interaction and reactivity. A severely over-simplified/qualitative approach is proposed based simply on approximate DFT adsorption energies.

Overall, this is a high quality study with excellent data quality, new ideas, and beautiful results. The sizing of the particles is clearly demonstrated. The presentation and quality of figures is excellent. All experimental aspects seem to be clearly reported and the supporting information are useful. Rather than one outstanding result, there seem to be several key strands of work reported here:

(A) the idea of adsorption effect in the collisions and an attempt to employ DFT to predict adsorption

(B) the observation of the AgO (2 electron) formation and the observation that this is particularly reliable for sizing

(C) the use of innovative equipment

(D) the actual sizing with good agreement

(E) the classification of different types of interactions

Each of these is good, but it is difficult to see the real individual breakthrough in these results. The DFT method is not quantitative and not easy to link to the experimental results. Electrolyte effects and charges on the surface must be really important and from the data I am not convinced that this simple DFT model is a crucial innovation.

The interesting observation of AgO (2 electron) is carefully described, but has been already been reported by the same authors before. This paper is not cited here: Correlated Anodic-Cathodic Nanocollision Events Reveal Redox Behaviors of Single Silver Nanoparticles By:Hafez, ME (Hafez, Mahmoud Elsayed)[1,2,3] ; Ma, H (Ma, Hui)[1,2] ; Peng, YY (Peng, Yue-Yi)[1,2] ; Ma, W (Ma, Wei)[1,2] ; Long, YT (Long, Yi-Tao)[1,2,4] JOURNAL OF PHYSICAL CHEMISTRY LETTERS Volume: 10 Issue: 12 Pages: 3276-3281 DOI: 10.1021/acs.jpcllett.9b01369 Published: JUN 20 2019

The use of innovative equipment is highlighted in the abstract, but not really reported in further detail.

The reported sizing in high quality is very good, but only confirms earlier work on sizing. I would like to see some more information about the production of these nanoparticles and the capping agents employed. Would capping affect the DFT results?

The classification of different types of interactions is useful, but there are already new results going beyond this simplified picture. The cited report [18] clearly shows additional effects from the ion formation causing propulsion of nanoparticles at surfaces, etc.

The use of nanoelectrodes in mechanistic diagnosis has been interesting and is may be worth also citing: Collision and Oxidation of Silver Nanoparticles on a Gold Nanoband Electrode By:Zhang, F (Zhang, Fan)[1] ; Edwards, MA (Edwards, Martin A.)[2] ; Hao, R (Hao, Rui)[1] ; White, HS (White, Henry S.)[2] ; Zhang, B (Zhang, Bo)[1] JOURNAL OF PHYSICAL CHEMISTRY C Volume: 121 Issue: 42 Pages: 23564-23573 DOI: 10.1021/acs.jpcc.7b08492 Published: OCT 26 2017.

The reference [13] by Ustaroz is in many ways very similar to the work reported here. So, somehow the innovation or the new break through needs to be better clarified.

The title "Exploring dynamic interaction of single nanoparticle at the interface for surface-confined electrochemical behavior" is very vague and doesn't indicate any particular achievement. What is the real achievement here?

Overall, this is a high quality study with very interesting results. This work should be published, but a revision might be a good idea to better clarify the key novelty and to better distinguish this work as outstanding and going beyond related work in the literature.

Reviewer #3:

Remarks to the Author:

In the manuscript a joint first-principle and experimental study of the interaction between nanoparticles (NPs) and electrode surface in impact experiments is performed with two kinds of particles (Ag and AgOx) and ultramicroelectrodes (carbon and gold). The different NP-electrode interaction (adsorption) predicted by DFT calculations correlates with different

spike shape, duration and charge, which is interpreted as the stronger restriction of the NP motion that promotes its quantitative electrolysis. This extends the size detection limit of electrochemical impact experiments.

The work is very well and clearly presented in text and figures, and the interest and impact of the topic (single-entity electrochemistry) is undeniable. Nevertheless, in my view there are some issues that should be improved or clarified before making a final decision on its suitability for publication in this very high-impact journal:

- The impact of the results. The overall main conclusion is somehow expected and described in the past, that is, the stronger the adsorption between NP and electrode is, the longer the collision time and the higher the particle electrooxidation. Also, it is not stated clearly the generality and applicability of the so-called "surface-confined nano-collision strategy" to other NP-electrode systems.

- As stated on page 15, "It is impossible to describe with certainty all the relevant interactions of NPs on an electrochemical interfaces". However, it is still worth discussing the main simplifications of the theoretical model and when these are expected to be relevant or not; thus, the following factors have been neglected in the simulations:

- o The electrical double layer of the electrode|solution interface. Was the applied potential in the experiments the potential of zero charge? Otherwise, were the results dependent on the value of the applied potential?

- o The capping agent of the nanoparticles, both in the calculations (neglected) and in the experiments (not indicated).

- o The presence of other species in solution, particularly of ions that can adsorb at the electrode|solution interface.

- On page 8, it is stated that when using the C ultramicroelectrode the particles undergo "consecutive multistep collisions": Is this a tentative explanation or is it supported by further evidence? On which basis is this discriminated from the possibility of multiple collisions of different particles? Was an analysis carried out to determine the window of time where there is very low probability of multiple collisions?

Other points:

- Page 9: Equation (2) is not a "diffusion equation" but the solution of a diffusion boundary value problem.

- Page 9: Was the formation of silver oxide confirmed by any method?

- Figure 1c: What do the error bars mean?

- Figure 3: Why does the electrochemical method underestimate the particle size with respect to TEM and DLS?

Response to reviewers:

Reviewer 1:

Comments:

This is a high quality paper, detailing a thorough experimental and simulation based approach to considering the oxidation of Ag nanoparticles during electrochemical collisions.

The data is well presented and analysed appropriately and inferences/interpretations are made based on literature, DFT simulation which are reasonable and logical.

This work is a valuable addition to a rapidly expanding field, and builds on the earlier body of literature nicely.

Q: I am happy to recommend publication in present form, albeit a reference to justify the applicability of a polar continuum model for solvent within the DFT calculations at this length scale would be welcomed in the text.

R: Thank you very much for your positive comments. We have added some references to justify the applicability of a polar continuum model for solvent within the DFT calculations as follows:

63. Tomasi, J., Mennucci, B. & Cammi, R. Quantum mechanical continuum solvation models. *Chem. Rev.* **105**, 2999-3094 (2005).

64. Gray, C. M., Saravanan, K., Wang, G. & Keith, J. A. Quantifying solvation energies at solid/liquid interfaces using continuum solvation methods. *Mol. Simulat.* **43**, 420-427 (2017).

On Page 15, we have changed “*To simulate the interface solution environment, the solvent effects were in combination with the polarizable continuum model.*” as “*To simulate the interface solution environment, the solvent effects were in combination with the **polarizable continuum model.***^{63,64}” in the revised Manuscript.

Reviewer 2:

Comments:

This study contributes to the "sizing" of silver nanoparticles as a particular model system in single entity electrochemistry. This work also contributes to the discovery of surface phenomena that contribute to the nanoparticle-surface interaction and reactivity. A severely

over-simplified/qualitative approach is proposed based simply on approximate DFT adsorption energies.

Overall, this is a high quality study with excellent data quality, new ideas, and beautiful results. The sizing of the particles is clearly demonstrated. The presentation and quality of figures is excellent. All experimental aspects seem to be clearly reported and the supporting information are useful. Rather than one outstanding result, there seem to be several key strands of work reported here:

(A) the idea of adsorption effect in the collisions and an attempt to employ DFT to predict adsorption

(B) the observation of the AgO (2 electron) formation and the observation that this is particularly reliable for sizing

(C) the use of innovative equipment

(D) the actual sizing with good agreement

(E) the classification of different types of interactions

Overall, this is a high quality study with very interesting results. This work should be published, but a revision might be a good idea to better clarify the key novelty and to better distinguish this work as outstanding and going beyond related work in the literature.

Q1: Each of these is good, but it is difficult to see the real individual breakthrough in these results. The DFT method is not quantitative and not easy to link to the experimental results. Electrolyte effects and charges on the surface must be really important and from the data. I am not convinced that this simple DFT model is a crucial innovation.

A1: Thank you very much for your valuable comments. In this work, we generalized scaling relations to **semi-quantitatively predict the adsorptive interaction trends of NPs on UME interfaces**. Based on the theoretical predictions, we explored the interaction-modulated

electrochemical oxidation behaviors of individual AgNPs. According to the reviewer's comments, we further examined the effects of electrolyte and charges in our DFT model, respectively.

1) Electrolyte effects

In this study, we investigated the electrochemical responses of individual AgNPs collisions at +600 mV vs Ag/AgCl (i) in neutral solution (25 mM PB, pH = 7.4) at a C UME; (ii) in neutral solution (25 mM PB, pH = 7.4) at a Au UME; and (iii) in alkaline media (15 mM PB and 10 mM NaOH, pH = 11.4) at a Au UME. Considering the presence of species in the solution, we calculated their adsorption energies of the possible species on the NPs surface and electrode interface by the first principle DFT calculations. As shown in Supplementary Table 4, the other species except for Na⁺ were found to have the weak affinity at the NP surface and electrode interface. For Na⁺ ion, we can see that the competitive adsorption energies of the NPs on the electrode interface are much more exothermic than that of Na⁺ on Ag, AgO_x and Au (*Na/Ag* vs *Ag/C*, *Na/AgO_x* vs *AgO_x/Au* and *Na/Au* and *Ag/Au* are -0.20 vs -0.91 eV, -2.16 vs -9.75 eV and -0.81 vs -5.98 eV, respectively). Moreover, together with the low concentration of the species (<<1 mol L⁻¹) in the solution, the coverage or probability of these species on the surface would expect to be very low. Therefore, the adsorption of the possible species in the solution on the NP surface and the electrode interface could be ignored in our DFT calculation.

Table S4. Calculated DFT adsorption energies of E_{ad} for the possible species in the solution on the NP surface and the electrode interface.

	E_{ad} (eV)					
	$H_2PO_4^-$	HPO_4^{2-}	Na^+	OH^-	Au	Ag
Ag	-0.02	0.51	-0.20	0.02	-5.98	/
AgO_x	2.14	4.37	-2.16	1.10	-9.75	/
Au	0.76	1.54	-0.81	0.86	/	-5.98
C	0.59	2.91	0.97	1.29	/	-0.91

2) Charge effects

Considering the inclusion of electrons on the electrode during the electrochemical processes, we introduced two kind of methods for correcting the charge effects in this DFT calculations.

Firstly, we employed a simple electrode potential-corrected energy method, which is widely used in the electrochemical reaction proposed by Nørskov,^{S19} to investigate the charge influence to adsorption energy. In addition to the normal adsorption energy obtained by standard DFT calculations, the adsorption energy change introduced by the change in the electrode potential can be realized through shifting the energy level by $-neU$, where n is the electron transfer number for a given reaction. Considering the same applied potential at +600 mV vs Ag/AgCl used in our electrochemical measurement, the electrode potential effect was normally ignored due to the same correction value.

Table S5. Calculated charge values (q) for the Ag/C, Ag/Au and AgO_x/Au systems with different electrode potential (U).

U (eV)	q (e)		
	Ag/C	Ag/Au	AgO _x /Au
-1.00	-0.20	-0.26	-0.19
-0.50	-0.09	-0.08	-0.10
0.00	0.00	0.00	0.00
0.50	0.10	0.06	0.10
1.00	0.18	0.12	0.20

On this basis, we further estimated the electrode potential dependent effect to the adsorption energy using the charge correction method.^{S20} In this correction, the electrode surface are assumed to be a plane-parallel capacitor, and then corrected the value by the energy of $1/2CU^2$. By adding the different charges on the electrode surface, we calculated the potential changes of the systems by the DFT calculations (Supplementary Table 5). By the slop of q - U relations, we estimated the

capacitances of the electrode surfaces ($C = \Delta q/\Delta U$, Supplementary Fig. 19). Clearly, we can see that the capacitances (C) are very similar, so are the correction values $1/2CU^2$ (which are 0.918, 0.932 and 0.945 eV for the Ag/C, Ag/Au and AgO_x/Au systems, respectively). Two kind of methods have both demonstrated that the correction values of the applied potential were usually approached to be similar, and thus charge effects were also ignored in our simulations.

Supplementary Fig. 19 Calibration curves of the calculated electrode potential (U) with different charge values (q) for the Ag/C, Ag/Au and AgO_x/Au systems. The line has been fitted by regression analysis.

S19. Nørskov, J. K. *et al.* Origin of the overpotential for oxygen reduction at a fuel-cell cathode. *J. Phys. Chem. B* **108**, 17886-17892 (2004).

S20. Skúlason, E. *et al.* Density functional theory calculations for the hydrogen evolution reaction in an electrochemical double layer on the Pt (111) electrode. *Phys. Chem. Chem. Phys.* **9**, 3241-3250 (2007).

On Page 15, we have added “Considering the negligible contribution, we ignore the effects of electrolyte (Supplementary Table 4), charge (Supplementary Table 5 and Supplementary Fig. 19) and capping agent (Supplementary Table 6) to theoretical energy calculations in this DFT model.” in the revised Manuscript.

On Page S4, we have added **Supplementary Table 4** and **Supplementary Table 5** in the revised Supplementary Information.

On Page S20, we have added **Supplementary Fig. 19** and **the above discussion** in the revised Supplementary Information.

Q2: The interesting observation of AgO (2 electron) is carefully described, but has been already been reported by the same authors before. This paper is not cited here: Correlated Anodic-Cathodic Nanocollision Events Reveal Redox Behaviors of Single Silver Nanoparticles By: Hafez, ME (Hafez, Mahmoud Elsayed)[1,2,3]; Ma, H (Ma, Hui)[1,2]; Peng, YY (Peng, Yue-Yi)[1,2]; Ma, W (Ma, Wei)[1,2]; Long, YT (Long, Yi-Tao)[1,2,4] JOURNAL OF PHYSICAL CHEMISTRY LETTERS Volume: 10 Issue: 12 Pages: 3276-3281 DOI: 10.1021/acs.jpcllett.9b01369 Published: JUN 20 2019

A2: We really apologize for this mistake. In the revised Manuscript, we have cited this paper.

On Page 8, we have changed “Considering the formation of silver oxide in alkaline media³⁸⁻⁴², we performed the electrochemical oxidation of 34 nm AgNP at a Au UME in a pH = 11.4 of an alkaline solution containing 15 mM PB and 10 mM NaOH.” as “Considering the formation of silver oxide in alkaline **media**⁴²⁻⁴⁷, we performed the electrochemical oxidation of 34 nm AgNP at a Au UME in a pH = 11.4 of an alkaline solution containing 15 mM PB and 10 mM NaOH.” in the revised Manuscript.

47. Hafez, M. E., Ma, H., Peng, Y.-Y., Ma, W. & Long, Y.-T. Correlated anodic–cathodic nanocollision events reveal redox behaviors of single silver nanoparticles. *J. Phys. Chem. Lett.* **10**, 3276–3281 (2019).

Q3: The use of innovative equipment is highlighted in the abstract, but not really reported in further detail.

A3: Thank you very much for your comment. With the development of new instruments and methodologies, the highly dynamic behaviors of NP at the liquid-solid interface have been studied. Over the past several decades, a variety of optical-based techniques and electron microscopies have been and are currently being investigated as means of identifying and characterizing NPs at the interface. For example, total internal reflection fluorescence microscopy was used to track the motion of individual polymer chains at a hydrophobic solid-liquid interface.^{R1} Alternatively, in situ electron microscope is a technique commonly used to directly monitor the real-time diffusion of NPs in solution at the liquid-solid interface.^{R2, R3}

Interestingly, single-entity electrochemical techniques that offer direct transduction to electron flux can enable studies of dynamic electrochemical processes of individual NPs at the electrode interface. In this study, we employed Axopatch 200B amplifier and DigiData 1550A low-noise data acquisition system (Molecular Devices, US) to recode the high-resolution electrochemical signals of individual AgNPs at a sampling rate of 100 kHz and an internal low-pass Bessel filter of 5 kHz. This means the time resolution of this instrument is 66 μ s, which is sufficient to measure the oxidation process of AgNPs (0.1 ~ 100 ms) in this system. Besides, the Axopatch 200B provides the lowest-noise single channel recording available. The open-circuit noise is 0.52 pA rms, when the Bessel filter was set as 5 kHz, which enabled accurate recording reaction electrochemical responses of single NPs with high signal/noise ration. To make a clear understanding of the electrochemical behavior of individual NPs, the combination of stochastic collision electrochemistry with the complementary techniques was introduced as a new tool for real-time analysis of the electrochemical reaction mechanism/kinetics of individual NPs. Recently, the simultaneous electrochemical measurement and scanned nanoprobe microscopies (e.g. scanning electrochemical microscope^{R4} and scanning electrochemical cell microscopy^{R5}) or optical-based techniques^{R6} have emerged to assist the identification of the electrochemical collision behaviors of individual NPs at an UME with sufficient spatial and temporal resolution.

- R1. Skaug, M. J., Mabry, J. N. & Schwartz, D. K. Single-molecule tracking of polymer surface diffusion. *J. Am. Chem. Soc.* **136**, 1327-1332 (2013).
- R2. Chee, S. W., Baraissov, Z., Loh, N. D., Matsudaira, P. T. & Mirsaidov, U. Desorption-mediated motion of nanoparticles at the liquid–solid interface. *J. Phys. Chem. C* **120**, 20462-20470 (2016).
- R3. Zheng, H., Claridge, S. A., Minor, A. M., Alivisatos, A. P. & Dahmen, U. Nanocrystal diffusion in a liquid thin film observed by in situ transmission electron microscopy. *Nano Lett.* **9**, 2460-2465 (2009).
- R4. Kim, J., Dick, J. E. & Bard, A. J. Advanced electrochemistry of individual metal clusters electrodeposited atom by atom to nanometer by nanometer. *Acc. Chem. Res.* **49**, 2587–2595 (2016).
- R5. Bentley, C. L., Kang, M. & Unwin, P. R. Nanoscale surface structure–activity in electrochemistry and electrocatalysis. *J. Am. Chem. Soc.* **141**, 2179–2193 (2019).
- R6. Wang, W. Imaging the chemical activity of single nanoparticles with optical microscopy. *Chem. Soc. Rev.* **47**, 2485–2508 (2018).

According to the reviewer’s comment, we have changed “Although the combination of stochastic collision electrochemistry with **the other complementary techniques** was introduced as a new tool for real-time analysis of the electrochemical reaction mechanism/kinetics of individual NPs,^{18,19} it remains an open and challenging issue to elucidate how nanoscale particles interact with an electrode interface to clearly understand the dynamic electrochemical processes of such particles.^{20–22}” as “Although the combination of stochastic collision electrochemistry with **scanned nanoprobe microscopies^{19,20} or optical-based techniques²¹** was introduced as a new tool for real-time analysis of the electrochemical reaction mechanism/kinetics of individual NPs **with high spatial and temporal resolution,^{22,23}** it remains an open and challenging issue to elucidate how nanoscale particles interact with an electrode interface to clearly understand the dynamic

electrochemical processes of such particles.²⁴⁻²⁶ in the introduction part of the revised Manuscript.

19. Kim, J., Dick, J. E. & Bard, A. J. Advanced electrochemistry of individual metal clusters electrodeposited atom by atom to nanometer by nanometer. *Acc. Chem. Res.* **49**, 2587–2595 (2016).
20. Bentley, C. L., Kang, M. & Unwin, P. R. Nanoscale surface structure–activity in electrochemistry and electrocatalysis. *J. Am. Chem. Soc.* **141**, 2179–2193 (2019).
21. Wang, W. Imaging the chemical activity of single nanoparticles with optical microscopy. *Chem. Soc. Rev.* **47**, 2485–2508 (2018).
22. Sundaresan, V., Monaghan, J. W. & Willets, K. A. Visualizing the effect of partial oxide formation on single silver nanoparticle electro-dissolution. *J. Phys. Chem. C* **122**, 3138–3145 (2018).
23. Robinson, D. A. & White, H. S. Electrochemical synthesis of individual Core@Shell and hollow Ag/Ag₂S nanoparticles. *Nano Lett.* **19**, 5612–5619 (2019).
24. Ma, W., Ma, H., Yang, Z.-Y. & Long, Y.-T. Single Ag nanoparticle electro-oxidation: Potential-dependent current traces and potential-independent electron transfer kinetic. *J. Phys. Chem. Lett.* **9**, 1429–1433 (2018).
25. Edwards, M. A. *et al.* Nanoscale electrochemical kinetics & dynamics: The challenges and opportunities of single-entity measurements. *Faraday Discuss.* **210**, 9–28 (2018).
26. Sun, L., Wang, W. & Chen, H.-Y. Dynamic nanoparticle-substrate contacts regulate multi-peak behavior of single silver nanoparticle collisions. *ChemElectroChem* **5**, 2995-2999 (2018).

On Page S1, we have changed “The amplifier's internal low-pass Bessel filter was set as 5 kHz. Data were acquired at a sampling rate of 100 kHz.” as “**In this study, we employed Axopatch 200B amplifier and DigiData 1550A low-noise data acquisition system (Molecular Devices, US) to recode the high-resolution electrochemical signals of individual AgNPs at a sampling rate of**”

100 kHz and an internal low-pass Bessel filter of 5 kHz.” in the revised Instrument part of Supplementary Information.

Q4: The reported sizing in high quality is very good, but only confirms earlier work on sizing. I would like to see some more information about the production of these nanoparticles and the capping agents employed. Would capping affect the DFT results?

A4: Thanks a lot for your concerns. In this work, citrate-capped AgNPs were used for the entire electrochemical experiment. We further investigated the capping agent effects of citrate ion to DFT results. According to the obtained results by the first principle DFT calculations, the adsorption energies of the citrate ion on the AgNP and AgO NP surface are -0.02 eV and 0.26 eV, respectively (Supplementary Table 6). Compared with the adsorption energies of NP on the electrode surface (Supplementary Table 2), the interactions between citrate ion and NP were significantly weak.

Moreover, due to the continuously refreshed particle surface during the electrochemical oxidation of AgNPs and the trace citrate ion in the solution, the probability of citrate ion back to NPs surface would expect to be very low. Taken together, we did not take the effect of the surface capping agent (citrate) into consideration for adsorption energy in this DFT calculation.

Table S6. Calculated DFT adsorption energies of E_{ad} for the citrate ion on the surface of AgNP and AgO NP.

E_{ad} (eV)	
citrate/Ag	citrate/AgO
-0.02	0.26

On Page 15, we have added “Considering the negligible contribution, we ignore the effects of electrolyte (Supplementary Table 4), charge (Supplementary Table 5 and Supplementary Fig.

19) and capping agent (Supplementary Table 6) to theoretical energy calculations in this DFT model.” in the revised Manuscript.

On Page S4, we have added **Supplementary Table 6** in the revised Supplementary Information.

On Page S22, we have added **the above discussion** in the revised Supplementary Information.

Q5: The classification of different types of interactions is useful, but there are already new results going beyond this simplified picture. The cited report [18] clearly shows additional effects from the ion formation causing propulsion of nanoparticles at surfaces, etc.

A5: Thanks for the reviewer’s comment. In the cited report [18], the temporal and spatial dependence of electrodissoolution of single localize AgNPs were investigated to study the heterogeneous surface oxide formation using *in situ* superlocalization optical microscopy. Moreover, the authors examined the particles shifting in position over time due to the motion on the surface induced by electro-osmotic or electrophoretic flow. At sufficiently positive potentials (≥ 1.2 V), these regions of local substrate potentials exert an electrostatic force on the negatively charged citrate-capped Ag NPs that overcomes the adhesion of the particle to the indium-tin oxide (ITO) surface, causing a shift in its position. For potentials < 1.2 V, the electrostatic force is not high enough to overcome adhesive forces, and thus $d < 10$ nm (e.g., the particles do not move when the positive potential is applied). Therefore, the electrostatic force between NP and electrode interface is also an important interaction at sufficiently positive potentials (≥ 1.2 V). **In this work, we investigated the electrochemical behaviors of individual AgNPs collisions at +600 mV vs Ag/AgCl. Thus, the electrostatic force was not the dominant factor in our system.**

[18] Sundaresan, V., Monaghan, J. W. & Willets, K. A. Visualizing the effect of partial oxide formation on single silver nanoparticle electrodissoolution. *J. Phys. Chem. C* **122**, 3138–3145 (2018).

On page 14, we have changed “Through this study, we have revealed the implicit role that the adsorptive interaction of NPs on the electrode interface plays in particle motion. It is impossible to describe with certainty all the relevant interactions of NPs on an electrochemical interface, but the current pockets of adsorption knowledge provide a conceptual framework to guide this exploration.” as “Through this study, we have revealed the implicit role that the adsorptive interaction of NPs on the electrode interface plays in particle motion. **In the previously obtained result, the electrostatic force is also an important interaction between AgNPs and electrode interface at sufficiently positive potentials (≥ 1.2 V).**”²² Therefore, it is impossible to describe with certainty all the relevant interactions of NPs on an electrochemical interface, but the current pockets of adsorption knowledge provide a conceptual framework to guide this exploration.” in the revised manuscript.

22. Sundaresan, V., Monaghan, J. W. & Willets, K. A. Visualizing the effect of partial oxide formation on single silver nanoparticle electrodisolution. *J. Phys. Chem. C* **122**, 3138–3145 (2018).

Q6: The use of nanoelectrodes in mechanistic diagnosis has been interesting and is may be worth also citing: Collision and Oxidation of Silver Nanoparticles on a Gold Nanoband Electrode By: Zhang, F (Zhang, Fan)[1]; Edwards, MA (Edwards, Martin A.)[2]; Hao, R (Hao, Rui)[1]; White, HS (White, Henry S.)[2]; Zhang, B (Zhang, Bo)[1] JOURNAL OF PHYSICAL CHEMISTRY C Volume: 121 Issue: 42 Pages: 23564-23573 DOI: 10.1021/acs.jpcc.7b08492 Published: OCT 26 2017.

A6: Thank you for your comment. The use of nanoelectrodes have many unique properties to measure ultrafast electron transfer process, particularly when the diffusion and capacitance were considered. The use of different nanoelectrodes has promoted the development of single-entity electrochemistry. Besides, the corresponding reference has now been cited in the revised Manuscript.

On Page 3, we have changed “Over the past several decades, electrochemical measurements of individual entities, including NPs, cells, molecules, and atoms, have been achieved conveniently with the use of stochastic collision measurement at a ultramicroelectrode (UME) or nanoelectrode.⁴⁻⁵” as “Over the past several decades, electrochemical measurements of individual entities, including NPs, cells, molecules, and atoms, have been achieved conveniently with the use of stochastic collision measurement at a ultramicroelectrode (UME) or nanoelectrode.⁴⁻⁶” in the revised Manuscript.

4. Quinn, B. M., van't Hof, P. G. & Lemay, S. G. Time-resolved electrochemical detection of discrete adsorption events. *J. Am. Chem. Soc.* **126**, 8360–8361 (2004).
5. Ma, H., Ma, W., Yang, Z., Ding, Z. & Long, Y.-T. Characterization of steady-state current at nanoelectrodes. *Acta Chim. Sin.* **75**, 1082 (2017).
6. Zhang, F., Edwards, M. A., Hao, R., White, H. S. & Zhang, B. Collision and oxidation of silver nanoparticles on a gold nanoband electrode. *J. Phys. Chem. C* **121**, 23564-23573 (2017).

Q7: The reference [13] by Ustaroz is in many ways very similar to the work reported here. So, somehow the innovation or the new break through needs to be better clarified.

A7: Thanks for the reviewer’s comment. The reference [13] by Ustaroz pioneered the application of scanning electrochemical cell microscopy (SECCM) for stochastic collision measurement of individual AgNPs in neutral solution of 50 mM NaNO₃. For 10 nm AgNPs, they are mostly dissolved in a single collision event. However, AgNPs of larger diameter ($d \geq 20$ nm) undergo sequential partial oxidative dissolution events, where a fraction of a NP is electrochemically oxidized, followed by the NP drifting away and back to the tunneling region before the next partial stripping event. Therefore, the authors demonstrated that **analysis of the charge consumed by single events (so-called “impact coulometry”)** could not be used as a general method to determine the size of colloidal AgNPs.

Considering the less wetting (smaller contact area) of the glass carbon (GC) substrate, GC gave lower background currents than Au as the collector electrode for SECCM cells of similar area. The authors investigated the comparative measurements of 100 nm AgNPs on both GC and Au collector electrodes. Similar to GC, the most prevalent types of stripping events on the Au collector electrode are short, sharp, high-current and long low-current, saw-tooth events. However, there are some subtle differences. Using a Au electrode, sharp collision events span for 5–15 ms, approximately twice the duration as on GC. From a proper analysis of current-time transients, the authors deduced that different collision events resulted from the physicochemical interactions between NPs and polarized surfaces. **However, this is only a preliminary insight.** As the author's statement, "whereas understanding the physical origin of such patterns requires further study, they serve to highlight the complexity of electrochemical dissolution phenomena via single NP impacts and the need for sub-ms time resolution (or, in the future, better) to properly study such processes." in the reference [13].

In fact, **our work mainly focused on the studies for exploring how to modulate the dynamic electrochemical behaviors of a single nanoparticle.** Here, we generalized scaling relations to predict NP-electrode interactions by examining the adsorption energy of NPs on an ultramicroelectrode interface by means of coordination number. Based on the theoretical predictions, we successfully localized the stochastic motion of particles in the vicinity of the electrode and modulated the oxidation process in AgO_x/Au system, producing the surface-confined current traces. Taking advantage of the surface-confined nano-collision strategy, we employed stochastic collision electrochemistry for the size measurements of AgNPs in mixed-sample systems. **Our method pushes the detection limit for stochastic collision measurements from the previous detection limit of ~30 nm diameter citrate-capped AgNPs in nitrate-based electrolyte and ~50 nm diameter of AgNPs in the presence of halide ions up to ~75 nm AgNPs in the present study. Through this study, we have revealed the implicit**

role that the adsorptive interaction of NPs on the electrode interface plays in particle motion.

[13] Ustarroz, J., Kang, M., Bullions, E. & Unwin, P. R. Impact and oxidation of single silver nanoparticles at electrode interfaces: one shot versus multiple events. *Chem. Sci.* **8**, 1841–1853 (2017).

On page 4, we have changed “Thus, adsorptive interactions play a key role in the physical motion of NPs at the electrode interface, yet clarifying the dependence of NP-electrode interactions and the dynamic nature of single-NP electrochemical behavior remains unexplored.” as “Thus, adsorptive interactions play a key role in the physical motion of NPs at the electrode interface. Recent electrochemical results provided a preliminary insight on the physicochemical interactions between NPs and electrode surfaces,¹⁴ yet clearly clarifying the dependence of NP-electrode interactions and the dynamic nature of single-NP electrochemical behavior remains unexplored.” in the revised Manuscript.

14. Ustarroz, J., Kang, M., Bullions, E. & Unwin, P. R. Impact and oxidation of single silver nanoparticles at electrode interfaces: one shot versus multiple events. *Chem. Sci.* **8**, 1841–1853 (2017).

Q8: The title "Exploring dynamic interaction of single nanoparticle at the interface for surface-confined electrochemical behavior" is very vague and doesn't indicate any particular achievement. What is the real achievement here?

A8: According to the reviewer’s comments, we have changed the title “Exploring dynamic interaction of single nanoparticle at the interface for surface-confined electrochemical behavior” as “Exploring dynamic interaction of single nanoparticle at interface for surface-confined electrochemical behavior and size measurement” in the revised Manuscript.

Reviewer 3:

Comments:

In the manuscript a joint first-principle and experimental study of the interaction between nanoparticles (NPs) and electrode surface in impact experiments is performed with two kinds of particles (Ag and AgOx) and ultramicroelectrodes (carbon and gold). The different NP-electrode interaction (adsorption) predicted by DFT calculations correlates with different spike shape, duration and charge, which is interpreted as the stronger restriction of the NP motion that promotes its quantitative electrolysis. This extends the size detection limit of electrochemical impact experiments.

The work is very well and clearly presented in text and figures, and the interest and impact of the topic (single-entity electrochemistry) is undeniable. Nevertheless, in my view there are some issues that should be improved or clarified before making a final decision on its suitability publication in this very high-impact journal:

Q1: The impact of the results. The overall main conclusion is somehow expected and described in the past, that is, the stronger the adsorption between NP and electrode is, the longer the collision time and the higher the particle electrooxidation. Also, it is not stated clearly the generality and

applicability of the so-called “surface-confined nano-collision strategy” to other NP-electrode systems.

A1: Thank you very much for your valuable comments. Surface-confined nano-collision strategy has the generality and applicability in single molecule detection system. For example, nanopore functions as confined space for electrochemical single molecule analysis. As a molecule is captured and transported through the confined space of the nanopore, the ions occupying the nanopore are excluded, which results in a detectable electrical signal from the transit blockade event.^{R7,R8} Because of the confined space of the nanopore, the interaction between target molecule and inner wall of the biological nanopore is significantly enhanced and induces a characteristically electrochemical signal. In the previous report, molecular dynamics simulations suggested dA₁₄ fully inserted in the pore lumen, resulting in the strong interaction between dA₁₄ and the aerolysin nanopore, which was demonstrated by the longer duration time observed for dA₁₄.^{R9} Therefore, surface-confined nano-collision strategy described herein is versatile and can be extended to understand single molecule electrochemical process occurring at nanoscale spaces.

R7. Ying, Y. L. & Long, Y. T. Single-molecule analysis in an electrochemical confined space. *Sci. China Chem.* **60**, 1187-1190 (2017).

R8. Cao, C. & Long, Y. T. Biological nanopores: confined spaces for electrochemical single-molecule analysis. *Acc. Chem. Res.* **51**, 331-341 (2018).

R9. Cao, C. *et al.* Mapping the sensing spots of aerolysin for single oligonucleotides analysis. *Nature Commun.* **9**, 2823 (2018).

On Page 14, we have added “**Lastly, surface-confined nano-collision strategy described herein is versatile and can be extended to understand single molecule electrochemical process occurring at nanoscale spaces.**” in the revised Manuscript.

Q2: As stated on page 15, “It is impossible to describe with certainty all the relevant interactions of NPs on an electrochemical interfaces”. However, it is still worth discussing the main

simplifications of the theoretical model and when these are expected to be relevant or not; thus, the following factors have been neglected in the simulations:

Q2-1: The electrical double layer of the electrode|solution interface. Was the applied potential in the experiments the potential of zero charge? Otherwise, were the results dependent on the value of the applied potential?

A2-1: Thank you very much for your valuable comments. Considering the inclusion of electrons on the electrode during the electrochemical processes, we introduced two kind of methods for correcting the adsorption energies as a function of the electrode potential in this DFT calculations.

Firstly, we employed a simple electrode potential-corrected energy method, which is widely used in the electrochemical reaction proposed by Nørskov,^{S19} to investigate the effect of the applied potential to adsorption energy. In addition to the normal adsorption energy obtained by standard DFT calculations, the adsorption energy change introduced by the change in the electrode potential can be realized through shifting the energy level by $-neU$, where n is the electron transfer number for a given reaction. Considering the same applied potential at +600 mV vs Ag/AgCl used in our electrochemical measurement, the electrode potential effect was normally ignored due to the same correction value.

Table S5. Calculated charge values (q) for the Ag/C, Ag/Au and AgO_x/Au systems with different electrode potential (U).

U (eV)	q (e)		
	Ag/C	Ag/Au	AgO _x /Au
-1.00	-0.20	-0.26	-0.19
-0.50	-0.09	-0.08	-0.10
0.00	0.00	0.00	0.00
0.50	0.10	0.06	0.10
1.00	0.18	0.12	0.20

On this basis, we further estimated the electrode potential dependent effect to the adsorption energy using the charge correction method.^{S20} In this correction, the electrode surface are assumed to be a plane-parallel capacitor, and then corrected the value by the energy of $1/2CU^2$. By adding the different charges on the electrode surface, we calculated the potential changes of the systems by the DFT calculations (Supplementary Table 5). By the slop of q - U relations, we estimated the capacitances of the electrode surfaces ($C = \Delta q/\Delta U$, Supplementary Fig. 19). Clearly, we can see that the capacitances (C) are very similar, so are the correction values $1/2CU^2$ (which are 0.918, 0.932 and 0.945 eV for the Ag/C, Ag/Au and AgO_x/Au systems, respectively). Two kind of methods have both demonstrated that the correction values of the applied potential were usually approached to be similar, and thus charge effects were also ignored in our simulations.

Supplementary Fig. 19 Calibration curves of the calculated electrode potential (U) with different charge values (q) for the Ag/C, Ag/Au and AgO_x/Au systems. The line has been fitted by regression analysis.

S19. Nørskov, J. K. *et al.* Origin of the overpotential for oxygen reduction at a fuel-cell cathode. *J. Phys. Chem. B* **108**, 17886-17892 (2004).

S20. Skúlason, E. *et al.* Density functional theory calculations for the hydrogen evolution reaction in an electrochemical double layer on the Pt (111) electrode. *Phys. Chem. Chem. Phys.* **9**, 3241-3250 (2007).

On Page 15, we have added “Considering the negligible contribution, we ignore the effects of electrolyte (Supplementary Table 4), charge (Supplementary Table 5 and Supplementary Fig. 19) and capping agent (Supplementary Table 6) to theoretical energy calculations in this DFT model.” in the revised Manuscript.

On Page S4, we have added **Supplementary Table 4** and **Supplementary Table 5** in the revised Supplementary Information.

On Page S20, we have added **Supplementary Fig. 19** and **the above discussion** in the revised Supplementary Information.

Q2-2: The capping agent of the nanoparticles, both in the calculations (neglected) and in the experiments (not indicated).

A2-2: Thanks a lot for your concerns. In this work, citrate-capped AgNPs were used for the entire electrochemical experiment. We further investigated the capping agent effects of citrate ion to DFT results. According to the obtained results by the first principle DFT calculations, the adsorption energies of the citrate ion on the AgNP and AgO NP surface are -0.02 eV and 0.26 eV, respectively (Supplementary Table 6). Compared with the adsorption energies of NP on the electrode surface (Supplementary Table 2), the interactions between citrate ion and NP were significantly weak.

Moreover, due to the continuously refreshed particle surface during the electrochemical oxidation of AgNPs and the trace citrate ion in the solution, the probability of citrate ion back to NPs surface would expect to be very low. Taken together, we did not take the effect of the surface capping agent (citrate) into consideration for adsorption energy in this DFT calculation.

Table S6. Calculated DFT adsorption energies of E_{ad} for the citrate ion on the surface of AgNP and AgO NP.

E_{ad} (eV)

citrate/Ag	citrate/AgO
-0.02	0.26

On Page 15, we have added “**Considering the negligible contribution, we ignore the effects of electrolyte (Supplementary Table 4), charge (Supplementary Table 5 and Supplementary Fig. 19) and capping agent (Supplementary Table 6) to theoretical energy calculations in this DFT model.**” in the revised Manuscript.

On Page S1, we have changed “Silver nanoparticles (AgNPs) with different average nominal diameters of 10, 18, 34, 55, 65, and 75 nm were obtained by nanoComposix, Inc. (San Diego, CA, USA).” as “**Citrate-capped silver nanoparticles (AgNPs) with different average nominal diameters of 10, 18, 34, 55, 65, and 75 nm were obtained by nanoComposix, Inc. (San Diego, CA, USA).**” In the revised Supplementary Information.

On Page S4, we have added **Supplementary Table 6** in the revised Supplementary Information.

On Page S21, we have added **the above discussion** in the revised Supplementary Information.

Q2-3: The presence of other species in solution, particularly of ions that can adsorb at the electrode|solution interface.

A2-3: Thanks for the reviewer’s comments. In this study, we investigated the electrochemical responses of individual AgNPs collisions at +600 mV vs Ag/AgCl (i) in neutral solution (25 mM PB, pH = 7.4) at a C UME; (ii) in neutral solution (25 mM PB, pH = 7.4) at a Au UME; and (iii) in alkaline media (15 mM PB and 10 mM NaOH, pH = 11.4) at a Au UME. Considering the presence of species in solution, we calculated their adsorption energies of the possible species on the NPs surface and electrode interface by the first principle DFT calculations. As shown in Supplementary Table 4, the other species except for Na⁺ were found to have the weak affinity at the NP surface and electrode interface. For Na⁺ ion, we can see that the competitive adsorption

energies of the NPs on the electrode interface are much more exothermic than that of Na^+ on Ag, AgO_x and Au (Na/Ag vs Ag/C , Na/AgO_x vs AgO_x/Au and Na/Au and Ag/Au are -0.20 vs -0.91 eV, -2.16 vs -9.75 eV and -0.81 vs -5.98 eV, respectively). Moreover, together with the low concentration of the species ($\ll 1 \text{ mol L}^{-1}$) in the solution, the coverage or probability of these species on the surface would expect to be very low. Therefore, the adsorption of the possible species in the solution on the NP surface and the electrode interface could also be ignored in our DFT calculation.

Table S4. Calculated DFT adsorption energies of E_{ad} for the possible species in the solution on the NP surface and the electrode interface.

	E_{ad} (eV)					
	H_2PO_4^-	HPO_4^{2-}	Na^+	OH^-	Au	Ag
Ag	-0.02	0.51	-0.20	0.02	-5.98	/
AgO_x	2.14	4.37	-2.16	1.10	-9.75	/
Au	0.76	1.54	-0.81	0.86	/	-5.98
C	0.59	2.91	0.97	1.29	/	-0.91

On Page 15, we have added “Considering the negligible contribution, we ignore the effects of electrolyte (Supplementary Table 4), charge (Supplementary Table 5 and Supplementary Fig. 19) and capping agent (Supplementary Table 6) to theoretical energy calculations in this DFT model.” in the revised Manuscript.

On Page S4, we have added **Supplementary Table 4** in the revised Supplementary Information.

On Page S21, we have added **the above discussion** in the revised Supplementary Information.

Q3: On page 8, it is stated that when using the C ultramicroelectrode the particles undergo “consecutive multistep collisions”: Is this a tentative explanation or is it supported by further

evidence? On which basis is this discriminated from the possibility of multiple collisions of different particles? Was an analysis carried out to determine the window of time where there is very low probability of multiple collisions?

A3: Thanks for your concerns. In single NP collision electrochemical measurements at a C UME, a spike with closely spaced clustering was clearly observed (Fig. 2a). We attributed the consecutive multistep collisions to one AgNP electrooxidation event, when the space time of these sequential peaks less than the theoretical interval time of two collision events (τ), which can be calculated by equation R1:

$$\tau = \frac{1}{4D_{NP}C_{NP}r_eN_A} \quad (\text{R1})$$

where C_{NP} is the concentration of AgNPs (37.8 pM), r_e is the radius of the Au UME, and N_A is Avogadro's number. D_{NP} is the diffusion coefficient of the AgNPs, which can be calculated using the Stokes-Einstein equation^{R10}:

$$D_{NP} = \frac{k_B T}{6\pi\eta r_{NP}} \quad (\text{R2})$$

where k_B is the Boltzmann constant ($1.38 \times 10^{-23} \text{ J K}^{-1}$), T is the absolute temperature (298 K), η is the solution viscosity ($8.94 \times 10^{-4} \text{ P s}$) and r_{NP} is the radius of AgNP (17 nm). The diffusion coefficient of the AgNPs according to equation R2 is $1.44 \times 10^{-11} \text{ m}^2 \text{ s}^{-1}$. As a result, the theoretical interval time of two collision events of individual AgNPs is 122 ms on a 12.5 μm diameter collecting electrode. Therefore, the theoretical results suggested that it was unlikely that multiple collisions of different particles occurred on a milli-second timescale, the experimental observation of clusters of close, consecutive current spikes can rather be attributed to the oxidation of a single AgNP *via* a series of stages, during each of which the NP is partially oxidized.

R10. Xiao, X., Fan, F. R. F., Zhou, J. & Bard, A. J. Current transients in single nanoparticle collision events. *J. Am. Chem. Soc.* **130**, 16669-16677 (2008).

Other points:

Q4: Page 9: Equation (2) is not a “diffusion equation” but the solution of a diffusion boundary value problem.

A4: We really apologize for this mistake. On Page 9, we have corrected description as “The maximum current (i_{\max}) of a single AgNP on the surface of the electrode can be described by **equation (2) for a sphere NP**” in the revised Manuscript.

Q5: Page 9: Was the formation of silver oxide confirmed by any method?

A5: Thanks for your concern. To confirm the $2e^-$ process, we carried out differential pulse voltammetry (DPV) measurements using a AgNP-modified Au electrode (diameter 2 mm) in alkaline media with pH = 11.4. Four anodic peaks, referred to A1–A4 in the DPV results, coincide with those reported in previous reports and are associated with the formation of Ag_2O and AgO between -200 mV and +800 mV vs Ag/AgCl wire (Supplementary Fig. 8). Moreover, the integrated charge of individual collision events as a function of applied potential (Supplementary Fig. 10) and the X-ray photoelectron spectroscopy (XPS) results for the electrochemically oxidized product (Supplementary Fig. 11) further demonstrated the formation of AgO at +600 mV vs Ag/AgCl wire. Therefore, a $2e^-$ oxidation occurred in the process of Ag oxidation when holding the potential of +600 mV vs Ag/AgCl wire in alkaline media.

According to the reviewer’s comments, we also supplemented the X-ray Diffraction (XRD) characterization of 34 nm AgNPs at +600 mV vs Ag/AgCl in alkaline media (15 mM PB and 10 mM NaOH, pH = 11.4) at the indium-tin oxide (ITO) electrode for 1 h. As shown in the revised Supplementary Fig. 11c, the diffraction peaks observed around $2\theta = 30-60^\circ$ can be indexed to AgO with JCPDS No. 74-1743, further demonstrating the formation of silver oxide in our system.

Supplementary Fig. 11 The XPS spectrum measured on the as-prepared silver sample. XPS survey spectrum (a) obtained from the oxide of AgNPs onto the gold substrate at the potential of +0.6 V and +0.4 V vs Ag/AgCl wire for 2 h in alkaline media (pH = 11.4), and the corresponding high-resolution XPS spectra of Ag3d (b). All XPS data were corrected for sample charging during X-ray irradiation using adventitious hydrocarbon referencing (C1s at 284.8 eV). (c) XRD pattern of 34 nm AgNPs on the surface of ITO electrode after anodic polarization for 1 h at +600 mV vs Ag/AgCl in alkaline media (15 mM PB and 10 mM NaOH). Source data are provided as a Source Data file.

On Page 10, we have changed “Additionally, the integrated charge of individual collision events as a function of applied potential (Supplementary Fig. 10) and the X-ray photoelectron spectroscopy (XPS) results for the electrochemically oxidized product (Supplementary Fig. 11) further demonstrated the formation of AgO at +600 mV vs Ag/AgCl wire.⁵⁰⁻⁵²” as “Additionally, the integrated charge of individual collision events as a function of applied potential (Supplementary Fig. 10), the X-ray photoelectron spectroscopy (XPS) and X-ray Diffraction (XRD) results for the electrochemically oxidized product (Supplementary Fig. 11) further demonstrated the formation of AgO at +600 mV vs Ag/AgCl wire.⁵⁰⁻⁵²” in the revised Manuscript.

On Page S11, we have changed “The oxidation products mainly contain the Ag₂O and a small amount Ag⁰ with a bias potential of +0.4 V, while major constituents of the sample at the potential of +0.6 V are the AgO.” as “Moreover, the oxidation products mainly contain the Ag₂O and a small amount Ag⁰ with a bias potential of +0.4 V, while major constituents of the sample at the potential of +0.6 V are the AgO. Additionally, we also used X-ray Diffraction (XRD) to

characterize the 34 nm AgNPs dropped indium-tin oxide (ITO) electrode after anodic polarization for 1 h at +600 mV vs Ag/AgCl in alkaline media (15 mM PB and 10 mM NaOH). The diffraction peaks observed around $2\theta = 30-60^\circ$ can be indexed to AgO with JCPDS No. 74-1743 (Supplementary Fig. 11c), further demonstrating the formation of silver oxide in our system.^{S14,S15} in the revised Supplementary Information.

S14. Waterhouse, G. I. N., Bowmaker, G. A. & Metson, J. B. The thermal decomposition of silver (I, III) oxide: A combined XRD, FT-IR and Raman spectroscopic study. *Mater. Lett.* **47**, 319–323 (2001).

S15. Behpour, A. S. M. Synthesis and characterization of AgO nanostructures by precipitation method and its photocatalyst application. *J. Mater. Sci. Mater. Electron.* **27**, 1191–1196 (2016).

Q6: Figure 1c: What do the error bars mean?

A6: Thank you very much for your valuable comments. Usually, the mean absolute error of adsorption energy using the DFT calculation with Perdew-Burke-Ernzerhof (PBEsol) relative to the experimental data is usually 0.2~0.5 eV, while the relative error of each self-consistent DFT calculation can usually be negligible.^{R11} In this work, the error bars are based on 0.2 eV from the DFT calculation with PBEsol relative to the experimental data.

After reconsideration by the reviewer's comment, we believed the error bars might be meaningless since 0.2 eV is rough estimated value. Therefore, **we deleted the error bar in the revised Fig. 1c.**

Fig. 1 Modeling NPs-electrode interactions. a Schematic illustration of interaction-modulated dynamic electrochemical behaviors of single NPs. b Optimized most stable adsorption configurations of Ag atoms and O atoms with different coordination environments on the C (002) basal plane and the Au (111) surface for DFT calculation. Gray ball, red ball, green ball/blue line and yellow ball correspond to C, O, Ag and Au atoms, respectively. c Scaling relations in adsorption energy of NPs on C and Au UME surfaces as a function of generalized coordination number. Data for Ag/C (\blacktriangle), Ag/Au (\blacksquare) and AgO_x/Au (\bullet) are provided, respectively.

R11. Brockherde, F., Vogt, L., Li, L., Tuckerman, M. E., Burke, K. & Müller, K. R. Bypassing the Kohn-Sham equations with machine learning. *Nature Commun.* **8**, 872 (2017).

Q7: Figure 3: Why does the electrochemical method underestimate the particle size with respect to TEM and DLS?

A7: Thank you very much for your comments. We consider this may be due to the bandwidth limitations of electrochemical instrument. In this study, we employed Axopatch 200B amplifier and DigiData 1550A low-noise data acquisition system (Molecular Devices, US) to recode the current signals of single AgNPs at a sampling rate of 100 kHz and an internal low-pass Bessel

filter of 5 kHz. This means the time resolution of this instrument is 66 μ s. Accordingly, events shorter than 66 μ s were invisible in the 100 kHz trace. Moreover, the noise in this system is about 20 pA, the current signals less than this threshold cannot be obtained accurately. As a result, size distribution from the electrochemical measurement underestimate the particle size with respect to TEM and DLS.

Reviewers' Comments:

Reviewer #2:

Remarks to the Author:

The revision process has addressed all points raised and the quality of this report has been improved. Publication is now recommended.

Reviewer #3:

Remarks to the Author:

The authors have made a considerable effort to address the questions stated in my previous report, which is to be thanked. However, I think that their response to three of the questions is not satisfactory:

Q1: My point about the so-called in the manuscript "surface-confined nano-collision strategy" is that the value of the conclusions is overrated with this term since the behaviour reported is specific of silver particles and the details about the possible extension to other particles (metallic, polymeric, macromolecules,...) are very vague. Hence, in my view, the current results and discussions would not enable the authors to state that a new strategy is described.

Q2-1: A key point is that, in the new calculations performed, the shift of the electronic energy level at the electrode is considered but not the electric potential that develops along the electrical double layer at the electrode|solution interface, which is potential dependent and can be expected to have a very relevant effect given that silver particles are covered by a negatively-charged capping agent (see, for example, ChemElectroChem, 2015, 2, 112–118).

Q3: Equation (R1) employed to discard multicollisions is the result of a continuum modelling of the system that does not capture its stochastic nature. A more rigorous treatment can be easily performed through the Poisson distribution (see Supporting Information in Nanoscale, 2019,11, 1720-1727).

According to all the above, the potential impact of the results is not well supported in the current form of the manuscript and some key aspects (specifically, the double layer effects) are not properly discussed. Hence, I cannot recommend the publication of this work in such a high impact journal.

Thank you very much for the valuable comments. We have made a careful revision according to the referees' comments. The revision details are listed as follows:

Reviewers' comments:

Reviewer #2 (Remarks to the Author):

The revision process has addressed all points raised and the quality of this report has been improved. Publication is now recommended.

R: Thank you very much for your positive comments.

Reviewer #3 (Remarks to the Author):

The authors have made a considerable effort to address the questions stated in my previous report, which is to be thanked. However, I think that their response to three of the questions is not satisfactory:

Q1: My point about the so-called in the manuscript "surface-confined nano-collision strategy" is that the value of the conclusions is overrated with this term since the behaviour reported is specific of silver particles and the details about the possible extension to other particles (metallic, polymeric, macromolecules,...) are very vague. Hence, in my view, the current results and discussions would not enable the authors to state that a new strategy is described.

R1: Thanks for the reviewer's comment. This work offers a better understanding of dynamic interaction of single AgNPs at the electrochemical interface and displays size measurement applications of AgNPs using stochastic collision electrochemistry. In this study, using first-principle density functional theory calculations, we studied the adsorption-energy scaling relations by examining AgNPs on two different UME surfaces. This allowed us to semi-quantitatively predict the adsorptive interaction effect of AgNPs on UME interfaces and helped in identifying the dynamic electrochemical nature of single AgNPs during the stochastic collision process. Based on the theoretical predictions, we explored the interaction-modulated electrochemical oxidation behavior of individual AgNPs, producing significantly distinguished current traces. By attempting to suppress the stochastic path of AgNPs and control their motion in an alkaline media, that is the surface-confined nano-collision strategy, we successfully realized the *in-situ* size discrimination of AgNP mixtures with different diameters of 10, 18, 34, 55, 65 and 75 nm using the stochastic collision electrochemistry. **Our results did demonstrate the possibility**

that the surface-confined nano-collision strategy could be used to accurately measure the size of AgNP mixtures in liquid suspensions *in-situ*.

Figure R1. Current-time responses of individual 3.3 nm MoS₂ QDs collisions on a C UME (diameter 7.0 μm) at -700 mV vs Ag/AgCl in 10 mM NaClO₄ solution containing different concentrations of HClO₄ from 25 to 150 mM. Left: Chronoamperometric profiles for the electrochemical catalytic proton reduction of individual MoS₂ QDs with the same scale bar. Right: Close-ups of the representative time-resolved current traces.

Recently, we have also made some preliminary research progress about surface-confined nano-collision strategy on electrochemical collision behaviours of individual MoS₂ QDs. As shown in Figure R1, significant current transients were observed for proton reduction of single MoS₂ QDs in various HClO₄ concentrations. As expected, a single peak was observed at the HClO₄ concentration of 25, 35 and 50 mM. Notably, a clear increasing trend of the maximum current was observed as the concentration increased, which attribute to the more hydrogen evolution with the increase of proton concentration. However, the catalytic current showed an opposite trend at the HClO₄ concentration from 75 to 150 mM. Interestingly, besides a single peak,

a long undulating staircase current trace appeared at the proton concentration of 75 mM. It is worth note that in the concentration range from 75 to 100 mM not only the probability of the undulating staircase current traces gradually increased, but also the duration of individual signals significantly elongated. In the previous report^{R1}, the H₂ concentration at the electrode surface from proton reduction exceeds the critical concentration for nucleation when the HClO₄ concentration is higher than to 60 mM. Therefore, we assumed that an undulating staircase current trace associated with a gaseous H₂ bubble formation. A sharp drop of current was ascribed to a gas-phase bubble that covers most of the active MoS₂ QDs surface, leaving a small portion of the MoS₂/solution surface for further proton reduction. **Importantly, individual MoS₂ QDs were confined at the surface of C UME due to the formation of nanobubble, resulting in the significantly long dwell time of MoS₂ QDs in a high proton concentration. This observation further demonstrated that the surface-confined nano-collision strategy described herein is versatile and can be extended to understand the hydrogen evolution reaction process of single MoS₂ QDs.** In the next step, we will continue to consider the generality and applicability of the surface-confined nano-collision strategy and explore the possible extension to other particles (metallic, polymeric, macromolecules, *etc.*) at the electrode interface.

R1. Zhou, J., Zu, Y. & Bard, A. J. Scanning electrochemical microscopy: Part 39. The proton/hydrogen mediator system and its application to the study of the electrocatalysis of hydrogen oxidation. *J. Electroanal. Chem.* **491**, 22-29 (2000).

On Page 14, we have changed “Lastly, surface-confined nano-collision strategy described herein is versatile and can be extended to understand single molecule electrochemical process occurring at nanoscale spaces.” as “In the next step, we will consider the generality and applicability of the surface-confined nano-collision strategy and explore the possible extension to other particles (metallic, polymeric, macromolecules, etc.) at the electrode interface.” in the revised Manuscript.

Q2-1: A key point is that, in the new calculations performed, the shift of the electronic energy level at the electrode is considered but not the electric potential that develops along the electrical double layer at the electrode|solution interface, which is potential dependent and can be expected to have a very relevant effect given that silver particles are covered by a negatively-charged capping agent (see, for example, ChemElectroChem, 2015, 2, 112–118).

R2-1: Thanks for the reviewer’s comment. In our previous response, we estimated the electrode potential dependent effect on the adsorption energy using the charge correction method.^{S21} In this correction, the electrode surface is assumed to be a plane-parallel capacitor, and then corrected the value by the energy of $1/2CU^2$. By adding the different charges on the electrode surface, we calculated the potential changes of the systems by the DFT calculations (Supplementary Table 5).

Table S5. Calculated charge values (q) for the Ag/C, Ag/Au and AgO_x/Au systems with different electrode potential (U).

U (eV)	q (e)		
	Ag/C	Ag/Au	AgO _x /Au
-1.00	-0.20	-0.26	-0.19
-0.50	-0.09	-0.08	-0.10
0.00	0.00	0.00	0.00
0.50	0.10	0.06	0.10
1.00	0.18	0.12	0.20

By the slop of q - U relations, we estimated the capacitances of the electrode surfaces ($C = \Delta q/\Delta U$, Supplementary Fig. 19). Clearly, we can see that the capacitances (C) are very similar. To make an equivalent comparison, the electrical potential at the same distance of NP on electrode surface was used to describe the correction value. Although the electric potential U drops sharply with the increasing distance X according to Poisson Boltzmann equation, the electric potential U is still identical at a certain distance X on the electrode|solution interface.^{S22} As a result, the correction values of the electric potential ($1/2CU^2$) were usually approached to be similar for the Ag/C, Ag/Au and AgO_x/Au systems. For example, at the electric potential of +600 mV vs Ag/AgCl (corresponding to the state that the NP adsorbed on the electrode surface at the distance $X = 0$), the correction values $1/2CU^2$ are 0.918, 0.932 and 0.945 eV for the Ag/C, Ag/Au and AgO_x/Au systems, respectively. Therefore, the potential dependent effect was negligible in our simulations due to the similar correction value.

Supplementary Fig. 19 Calibration curves of the calculated electrode potential (U) with different charge values (q) for the Ag/C, Ag/Au and AgO_x/Au systems. The line has been fitted by regression analysis.

On Page S23, we have changed “Clearly, we can see that the capacitances (C) are very similar, so are the correction values $1/2CU^2$ (which are 0.925, 0.932 and 0.949 eV for the Ag/Au, AgO_x/Au and Ag/C systems, respectively). Two kind of methods have both demonstrated that the correction values of the applied potential were usually approached to be similar, and thus charge effects were also ignored in our simulations.” as “Clearly, we can see that the capacitances (C) are very similar. To make an equivalent comparison, the electrical potential at the same distance of NP on the electrode surface was used to describe the correction value. Although the electric potential U drops sharply with the increasing distance X according to Poisson Boltzmann equation, the electric potential U is still identical at a certain distance X on the electrode/solution interface.²² As a result, the correction values of the electric potential ($1/2CU^2$) were usually approached to be similar for the Ag/C, Ag/Au and AgO_x/Au systems. For example, at the electric potential of +600 mV vs Ag/AgCl (corresponding to the state that the NP adsorbed on the electrode surface at the distance $X = 0$), the correction values $1/2CU^2$ are 0.918, 0.932 and 0.945 eV for the Ag/C, Ag/Au and AgO_x/Au systems, respectively. Therefore, the potential dependent effect was negligible in our simulations due to the similar correction value. Two kinds of methods have both demonstrated that the correction values of the applied potential were usually approached to be similar, and thus charge effects were also ignored in our simulations.” in the revised Supplementary Information.

- S21. Skúlason, E. *et al.* Density functional theory calculations for the hydrogen evolution reaction in an electrochemical double layer on the Pt (111) electrode. *Phys. Chem. Chem. Phys.* **9**, 3241-3250 (2007).
- S22. Tschulik, K., Cheng, W., Batchelor-McAuley, C., Murphy, S., Omanović, D. & Compton, R. G. Non-invasive probing of nanoparticle electrostatics. *ChemElectroChem* **2**, 112-118 (2015).

Q3: Equation (R1) employed to discard multicollisions is the result of a continuum modelling of the system that does not capture its stochastic nature. A more rigorous treatment can be easily performed through the Poisson distribution (see Supporting Information in *Nanoscale*, 2019, 11, 1720-1727).

A3: Thank you very much for your valuable comments. In single NP collision electrochemical measurements at a C UME, a spike with closely spaced clustering was clearly observed (Fig. 2a). As for the experimentally measured current trace, two situations may occur during the electrochemical process. First, the consecutive multistep collisions attributed to one AgNP electrooxidation event. Second, there is the simultaneous collisions of two or more particles with the electrode and being detected at the same time. To clearly understand the stochastic collision process of individual NPs, we employed Poisson distribution treatment to statistically study the datasets of collision signals. In this theoretical model, we defined an interval time window of two collision events (t) during which only one or zero single NP can collide at the C UME surface with 99% confidence. That is, there is a 99% collision probability of up to one AgNP that occur within the time interval. This indicates that the experimentally observed multi-spikes within this time interval window are clustered as the multi-collisions of the same AgNP with the CUME. Under the assumption of independent collision events, the time interval window can be quantified by the probability theory of a Poisson distribution as follows by equation S1^{S1}:

$$P_n(t) = \frac{(\lambda t)^n}{n!} e^{-\lambda t} \quad (\text{S1})$$

where P is the probability and n is the number of collision events in a certain time interval t . λ is the average occurrence rate of such events, which can be calculated using a steady-state diffusion flux of the AgNPs to the C UME, J (s^{-1}), by equation S2:

$$J = 4D_{NP}C_{NP}r_{elec}N_A \quad (\text{S2})$$

where C_{NP} is the concentration of AgNPs (37.8 pM), N_A is Avogadro's constant (6.02×10^{23} mol⁻¹), and r_{elec} is the radius of the C UME (3.5 μm). The diffusion coefficient of an AgNP (D_{NP}) can be determined from the Stokes-Einstein equation, equation S3,

$$D_{NP} = \frac{k_B T}{6\pi\eta r_{NP}} \quad (S3)$$

where k_B is the Boltzmann constant (1.38×10^{-23} J K⁻¹), T is the absolute temperature (298 K), η is the solution viscosity (8.94×10^{-4} P s) and r_{NP} is the radius of AgNP (17 nm). The diffusion coefficient of the AgNPs according to equation S3 is 1.44×10^{-11} m² s⁻¹. Herein, the calculated flux J is the average occurrence rate λ in the Poisson distribution equation. As a result, the theoretical interval time of two collision events of individual AgNPs is 32 ms at a 7.0 μm diameter collecting electrode. Therefore, the theoretical results suggested that it was unlikely that multiple collisions of different particles occurred on a millisecond timescale, the experimental observation of clusters of close, consecutive current spikes can rather be attributed to the oxidation of a single AgNP via a series of stages, during each of which the NP is partially oxidized.

S1. Xie, R., Batchelor-McAuley, C., Young, N. P., Compton, R. G. Electrochemical impacts complement light scattering techniques for in situ nanoparticle sizing. *Nanoscale* **11**, 1720-1727 (2019).

On Page 8, we have changed "*Because consecutive multistep collision behaviors occur via a continuous-time random walk mechanism, the electrooxidation of single AgNPs experiences a longer timescale in this case.*" as "*Poisson distribution model demonstrates the consecutive current spikes can rather be attributed to the oxidation of a single AgNP via a series of stages, during each of which the NP is partially oxidized (see details in Supplementary Information); thus the electrooxidation of single AgNPs experiences a longer timescale in this case.*" in the revised Manuscript.

On Page S2, we have added **the above discussion** in the revised Supplementary Information.

Reviewers' Comments:

Reviewer #3:

Remarks to the Author:

The authors have addressed all the questions stated in my previous reports. In my view, now the strong and less strong points of the model are (either explicitly or implicitly) indicated to the reader. Then, I recommend the publication of this work.

Just a minor comment: the reference electrode in reference 22 (Pt wire) is different from that used in this work (Ag/AgCl). Then, the value of 1.2V mentioned in line 295 should either be converted to the Ag/AgCl scale or be referred to Pt wire.

-----**Response Letter from Authors**-----

Thank you very much for the valuable comments. We have made a careful revision according to the referees' comments. The revision details are listed as follows:

Reviewers' comments:

Reviewer #3 (Remarks to the Author):

The authors have addressed all the questions stated in my previous reports. In my view, now the strong and less strong points of the model are (either explicitly or implicitly) indicated to the reader. Then, I recommend the publication of this work.

Q1: Just a minor comment: the reference electrode in reference 22 (Pt wire) is different from that used in this work (Ag/AgCl). Then, the value of 1.2V mentioned in line 295 should either be converted to the Ag/AgCl scale or be referred to Pt wire.

R1: Thanks for the reviewer's comment. According to the reviewer's suggestion, we have characterized the potentials of several common reference electrodes in 1 mM $K_3Fe(CN)_6$ solution.

Figure R1. Potential conversion. Cyclic voltammograms in 1 mM $K_3Fe(CN)_6$ containing 0.1M KNO_3 for 2 mm Au electrode with different reference electrodes of Pt wire (black line), Ag/AgCl wire (red line) and SCE (blue line). The scan rate is 100 mV s^{-1} .

As shown in Figure R1, the potential difference between Ag/AgCl wire as a reference electrode and Pt wire as a reference electrode is + 0.19 V. In the previously obtained result, the electrostatic force is an important interaction between AgNPs and electrode interface at sufficiently positive potentials ($\geq +1.2 \text{ V vs Pt wire}$).^{R1} The converted potential to the Ag/AgCl scale is +1.39 V, which

is much higher than the applied potentials of +0.6 V vs Ag/AgCl wire in our system. Therefore, the electrostatic force was not the dominant factor in this study.

R1. Sundaresan, V., Monaghan, J. W. & Willets, K. A. Visualizing the effect of partial oxide formation on single silver nanoparticle electrodisolution. *J. Phys. Chem. C* **122**, 3138–3145 (2018).

On page 14, we have changed “*In the previously obtained result, the electrostatic force is also an important interaction between AgNPs and electrode interface at sufficiently positive potentials ($\geq +1.2$ V).*”²²” to “*In the previously obtained result, the electrostatic force is also an important interaction between AgNPs and electrode interface at sufficiently positive potentials ($\geq +1.2$ V vs Pt wire).*”²²” in the revised Manuscript.